# System characterization of a human-sized 3D real-time magnetic particle imaging scanner for cerebral applications
Florian Thieben [1,2,5] ✉, Fynn Foerger[1,2,5], Fabian Mohn [1,2,5], Niklas Hackelberg [1,2], Marija Boberg [1,2], Jan-Philipp Scheel [3,4], Martin Möddel [1,2], Matthias Graeser[3,4] & Tobias Knopp [1,2,3]

Since the initial patent in 2001, the Magnetic Particle Imaging community has endeavored to develop a human-applicable Magnetic Particle Imaging scanner, incorporating contributions from various research fields. Here we present an improved head-sized Magnetic Particle Imaging scanner with low power consumption, operated by open-source software and characterize it with an emphasis on human safety. The focus is on the evaluation of the technical components and on phantom experiments for brain perfusion. We achieved 3D single- and multi-contrast imaging at 4 Hz frame rate. The system characterization includes sensitivity, resolution, perfusion and multi-contrast experiments as well as field measurements and sequence analysis. Images were acquired with a clinically approved tracer and within human peripheral nerve stimulation thresholds. This advanced scanner holds potential as a tomographic imager for diagnosing conditions such as ischemic stroke (different stages) or intracranial hemorrhage in environments lacking electromagnetic shielding, such as the intensive care unit.

Human-scale magnetic particle imaging (MPI) has the potential to provide background- and radiation-free tomographic images, at high temporal resolution. In comparison, computed tomography (CT) provides high-resolution images at the cost of radiation exposure and is ill-suited for reoccurring or long-term monitoring, whereas Magnetic Resonance Imaging (MRI) suffers from limited patient accessibility and lengthy scan durations. Both modalities come as large and fixed systems, which makes it challenging to use them in an intensive care unit (ICU). The main challenge is that patients usually need to be transported, which is not without risks and requires considerable preparation[1]. In addition, patients are highly encapsulated in MRI systems, making it difficult to monitor their condition[2]. These are also the reasons why portable devices such as chest radiographs and ultrasounds are mainstay imaging modalities in ICUs[3]. Recently, portable MRI systems with low $B_0$ fields proved the potential of bedside brain imaging devices in the ICU[4,5]. MPI also has great potential in this direction, due to its size and compatibility with the ICU environment. MPI brain images could be acquired inside the ICU, directly at the patient's bed, reducing the workload for medical staff, avoiding transport of the patient, and shortening the time of treatment decisions. As a quantitative and tracer-based imaging modality, MPI is able to visualize the structure of larger blood vessels or quantify tissue perfusion with high temporal resolution, among other diagnostic and therapeutic applications[6]. The potential application in ICUs targets neurovascular diseases like ischemic stroke, intracranial hemorrhage, and traumatic brain injury that require immediate attention and post-treatment monitoring to evaluate the procedure. More than 17 million cases occur each year worldwide and are a leading cause of death and disability[7,8], motivating further research and development.

Currently, MPI is in the process of upscaling[9] the preclinical (small rodent) bore size[10,11] to match human proportions like the head[8,12,13] or extremities[14]. Images are acquired with high spatiotemporal resolution, providing background-free contrast information, based on the nonlinear response of magnetic nanoparticles (MNPs)[15] with high sensitivity[16,17]. Various combined magnetic fields in the low kHz range allow the spatial selection and detection of a tracer, such as the established contrast agent Ferucarbotran from MRI, with up to 46 volumes per second[10]. MPI was shown to be capable of hyperthermia treatment[18,19], stem cell labeling[20], detection of lymph node metastatsis[21], gut bleeding[22] and lung perfusion imaging[23] in murine models, cancer detection[24], as well as being useful in

¹Section for Biomedical Imaging, University Medical Center Hamburg-Eppendorf, Hamburg, Germany. ²Institute for Biomedical Imaging, Hamburg University of Technology, Hamburg, Germany. ³Fraunhofer IMTE, Fraunhofer Research Institution for Individualized and Cell-based Medical Engineering, Lübeck, Germany. ⁴Institute of Medical Engineering, University of Lübeck, Lübeck, Germany. ⁵These authors contributed equally: Florian Thieben, Fynn Foerger, Fabian Mohn. ✉e-mail: f.thieben@uke.de

interventional applications that involve guiding catheters[25,26] and stent positioning[27]. In addition, the MPI tracer can be used as a micro probe for several external and internal parameters that change the signal response, like the carrier medium viscosity[28] and temperature[29]. Furthermore, particle properties like binding state[30], the particle core-size[31,32], or the orientation[33], can be derived and visualized using multi-contrast imaging.

One objective of upscaling MPI scanners is to investigate clinical utility by evaluating the power required to generate the magnetic fields and by assessing the realistic resolution and sensitivity of real-time imaging at this scale. A major challenge is to minimize the scanners power consumption, especially of the selection field (SF)[34], as well as ensuring patient safety in proximity to high-power components. Our approach encompasses these conditions and attempts to combine standard-socket power supply, an unshielded environment and to meet local medical device safety regulations.

In this paper, we describe a human-sized MPI system for brain applications, and verify its functionality in several experiments using a clinically approved tracer[35]. The general scanner concept is based on Graeser et al.[8], although many hardware components were replaced, improved, or re-developed for this version to increase instrumental safety with a focus on future human trials. A major improvement is the realization of 3D imaging by using 2D excitation from two orthogonal drive-field coils (DFCs) and a slow shift of the dynamic selection field to achieve volumetric sampling at 4 Hz. In this work, we describe and analyze the full system design and implementation, including our excitation fields and the measured system matrix, as well as characterize the overall performance of the brain scanner with resolution-, sensitivity-, perfusion-, and multi-contrast experiments.

Our work represents a substantial step towards the clinical application of MPI and may pave the way for monitoring neurovascular diseases within the ICU. The scanner system described is suited for end users, with an adaptable and interactive graphical user interface (GUI), an open-source reconstruction framework, and redundant safety mechanisms, that facilitate performing MPI scans and the live inspection of results. We elaborate on all system components, explain how the magnetic fields are generated, provide insight into our custom signal receive chain, and characterize the imaging performance using the system matrix approach.

## Results
### System overview

The presented MPI brain scanner is captured in an image in Fig. 1a, and a schematic block diagram is depicted in Fig. 1b. The scanner can be classified into four main parts: operational control (subsubsection "Operational control"), field generation (subsubsections "Field generation and reception, Selection-field generation, and Imaging sequence"), signal reception (sub-subsection "Signal reception"), and data processing (subsection "Data processing"), which are briefly introduced in this section for an overview.

The operational control is tasked with coordinating signal generation (digital-to-analog converter (DAC)) and reception (analog-to-digital converter (ADC)). The system devices are driven and coordinated by a collection of open-source software[36,37]. The combined software stack realizes a framework, which is adaptable and scalable to many different MPI applications. The framework accepts user inputs via a command line interface, and, more conveniently, via a GUI (https://github.com/MagneticParticleImaging/MPIUI.jl), that allows different types of measurements to be started, paused, aborted, or stopped. Also, it enables live displaying and analyzing measurement data and controlling and monitoring system devices such as temperature sensors or robots. The acquired MPI signals are stored in the open MPI data format (MDF)[38]. Another part of the operation control is a surveillance unit (SU) for monitoring, based on the micro-controller board *Arduino Mega 2560 Rev3*. It processes sensor data, controls the state of signal relays, and communicates via analog pins with amplifiers, temperature units, and control units, as well as serial communication with the operational software. Some functions, status information and relays are routed to a hardware user console, placed with the human operator, allowing interaction via buttons to activate or deactivate key parts of the system at will.

Field generation is initiated by four DACs, which generate two drive-field (DF) signals and two signals for the dynamic selection field. The DF signal of each channel is connected to a floating transmission chain with a 5th-order band-pass filter that powers the drive-field generator (DFG) via an inductive coupling network (ICN). The DFG is responsible for 2D field excitation in the $xz$-plane (5, 4 mT) and forms a high-quality-factor resonator that is designed to carry high currents at low voltage (referred to as high current resonator (HCR)). The ICN is a toroidal transformer that serves three purposes, high current gain, circuit symmetry, and floating potentials to increase human safety. The dynamic selection field is generated by the selection-field generator (SFG) with two coils mounted on an iron yoke inside a copper cabin, identical to this part of the setup in ref. 8. Identical 10 A coil currents generate a 0.24 Tm$^{-1}$ gradient field with a field-free-point (FFP) in the center of the field of view (FOV). By varying these currents, the FFP is moved along the $y$-direction to create a large nominal 3D FOV ($84 \times 85 \times 67$ mm$^3$). The dynamic selection-field current waveform can be chosen to be sinusoidal, or in our case, triangular, for a constant shifting motion with a 4 Hz imaging sequence. This principle is shown in Fig. 1c and explained along with all mentioned fields in more detail in the subsection "Field generation and reception".

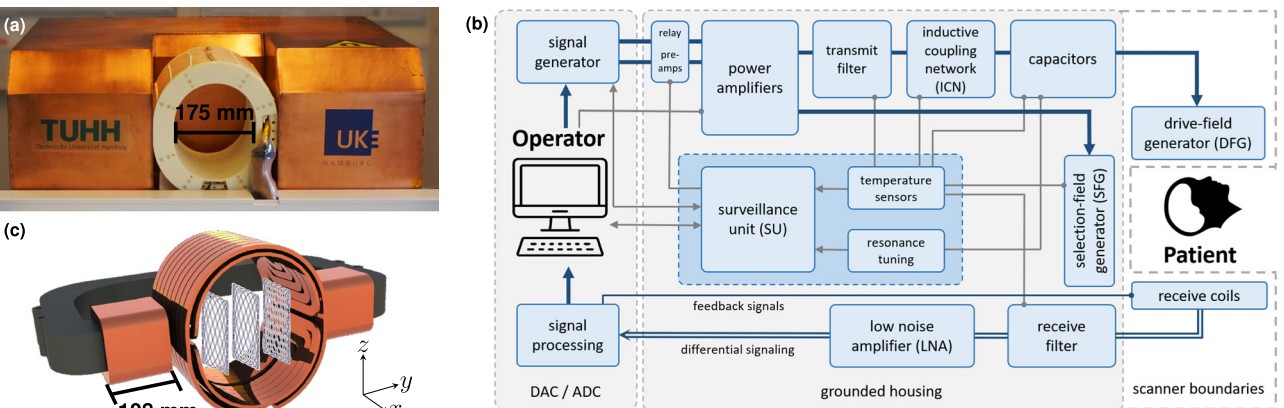

**Fig. 1 | General setup of the Magnetic Particle Imaging (MPI) brain scanner. a** Front view photo with the drive-field generator (DFG) in a white 3D-printed housing. **b** Schematic overview of the entire scanner using a flowchart that illustrates the key system components and their interactions. The digital-to-analog converter (DAC) and analog-to-digital converter (ADC) for signal generation and acquisition are realized by a stack of three RedPitaya STEMlab 125-14. **c** Rendering of the DFG, including the selection-field generator (SFG), which is an iron yoke that creates the dynamic selection field. Individual closed (ideal) 2D Lissajous trajectories are visualized in the $xz$-plane, that are shifted in $y$-direction.

Signal reception is realized with a gradiometric receive coil for feed-through suppression in the $x$-direction and a saddle coil in the $y$-direction. Given that both DFCs are orthogonal to the $y$-direction, this channel is not required to suppress high feedthrough signals by gradiometric turns. Both receive coils are connected to a symmetric fourth-order band-stop filter, transferred to single-ended signals via a balun transformer matching a single-ended custom low noise amplifier (LNA). Signals are then transferred differentially towards the ADCs, which are configured with a receive bandwidth of 976.56 kHz. Furthermore, two more signals are passed to the ADCs for feedback during the control step at the beginning of measurements, that automatically scales the transmit channel currents to match the desired DF strengths.

Data processing describes image reconstruction and further processing steps in the case of perfusion images. For image reconstruction, the system matrix approach is used, implemented by the open-source MPI reconstruction framework[39]. The measured data can be interpreted as multi-patch data corresponding to the selection-field shifts, or as a 3D single-patch dataset. We use a Kaczmarz-solver, system matrix over-gridding (interpolation), $L^2$-as well as $L^1$-regularization, background subtraction, and frequency selection to obtain images (see subsubsection "Image reconstruction" for details). These are presented for different experiments with individually tuned reconstruction parameters in the subsection "System characterization". Perfusion images are calculated based on reconstructed 3D volumes that are filtered and processed to obtain the time-to-peak (TTP), mean-transit-time (MTT), relative cerebral-blood flow (rCBF), and relative cerebral-blood volume (rCBV). The definition and details on the implementation can be found in the subsubsection "Perfusion image calculation".

## System characterization

To evaluate the performance of the developed MPI brain scanner and its characteristics, several experiments were conducted. We start with a low-level evaluation of the scanner and first study the generated drive field and selection field by using appropriate field measurements. In the next step, we perform an analysis of the acquired system matrix $S$, which allows to derive and predict the imaging performance, independent of specific phantoms. Then, the imaging performance is analyzed at the phantom level using simple sensitivity and resolution phantoms and later also using an application-relevant dynamic perfusion phantom. Finally, the suitability of the scanner for multi-contrast imaging is demonstrated.

**Field analysis.** The image quality of an MPI scanner is closely linked to the FFP trajectory, which in turn depends on the homogeneity of the drive and selection field. For the presented MPI scanner, the DFG is located in close proximity to the copper shielding of the selection-field generator (see Fig. 1). Hence, the drive field generates eddy currents inside the copper shielding, which in turn influences the field profile of the DFCs. For the selection-field coils, the cross-section area is relatively small compared to the distance between the coils, which leads to field inhomogeneities.

To obtain the actual imaging trajectory, the field profiles of the drive and selection fields can be represented as a spherical harmonic expansion by measuring a few points on a sphere rather than using a Cartesian grid interpolation, as reported by ref. 40. A 1D transmit sequence (0.248 s) at a defined DF amplitude of 5 and 4 mT was used to measure the $x$- and $z$-drive field. The dynamic magnetic fields were determined using a calibration robot with a mounted custom 3D coil sensor, with 86 measurement positions of a spherical 12-design[41,42]. Due to the $x$-receive coil turns at the front of the DFG, the FOV center and, thus the sphere center are shifted by 23 mm in $x$-direction from the geometric DFG center. In addition, these measurements were used to determine the fundamental total harmonic distortion (THD) of the drive fields[43]. The amplified and filtered DF signal induced into the coil sensor exhibits a THD of 0.0669% and 0.127% for the $x$- and $z$-channel, respectively. Regarding the selection field, the field of each coil was examined individually. Due to soft-iron-induced saturation

behavior, a list of 2, 4, 6, 8, 10, 12, and 14 A DC currents was set to measure the magnetic field at 36 positions of a spherical eight-design[42] using a three-axis high-sensitivity Hall-effect sensor with a three-channel gaussmeter (Model 460, Lake Shore Cryotronics, Inc., USA). This measurement allowed the adjustment of the relationship between current, gradient strength, and FFP position[40]. In Fig. 2, the field profiles for the $x$- and $z$-drive fields are shown in the top rows. Field inhomogeneities become stronger towards the edges of the FOV.

The selection field for identical 10 A currents in both coils is shown in the third row of Fig. 2. The largest gradient is observed in the $y$-direction, reaching 0.24 Tm$^{-1}$. In comparison, the gradients in the $x$- and $z$-direction are half as strong, measuring $-0.12$ Tm$^{-1}$. In the bottom row of Fig. 2, the 2D Lissajous trajectory is shown in the $xz$-plane for defined selection-field offsets of patches 3, 22, and 33. The trajectory is sampled by marking the FFP after the superposition of all three magnetic fields for certain time points. For visualization, the density of the trajectory is adapted by changing the frequency ratio to $\frac{16}{15}$. In the background, the selection field of each patch is shown in the $xz$-plane. From the measurements, the calculated 3D DF FOV spans a volume of $83 \times 80 \times 73$ mm$^3$.

**System matrix analysis.** In the next step, the imaging performance of the scanner is analyzed by studying a measured system matrix, which will later also be used for image reconstruction. The system matrix was acquired using a robot-based approach with a cubic 200 μL $\delta$-sample filled with *Resotran* (b.e.imaging GmbH, Germany) in a concentration of 8.5 mg$_{Fe}$mL$^{-1}$ (152 mol L$^{-1}$) to prevent magnetic dipole-dipole interactions[44]. The sample was shifted to $15 \times 15 \times 11$ positions covering a volume of $140 \times 110 \times 100$ mm$^3$, and at each position, one full 3D sequence (0.248 s) was applied. For later background subtraction and SNR analysis, 12 empty measurements after each $xy$-plane were performed by moving the sample with the calibration robot outside the scanner bore. During the acquisition, the DF feedback was tracked and the observed amplitudes and phases showed a standard deviation of below 0.4% over the 2491 measurements. All signals are considered in frequency space, which is common in MPI since it allows for direct filtering of interfering signals like the signal induced by the drive field.

The measured system matrix can be interpreted in two different ways. First, since the FFP movement induced by the selection-field generator is very slow, the data can be interpreted as a multi-patch dataset, where the 2D Lissajous trajectory ($xz$-plane) is slowly shifted to $M_y = 85$ positions along the $y$-axis. One can thus expect that the system-matrix patterns are just shifted in space, which was shown for idealized magnetic fields by refs. 45, 46. The measured system matrix considering this multi-patch processing is illustrated in Fig. 3. Shown are frequency patterns i.e., system matrix rows reshaped on the 3D grid at frequencies $f_k^{MP} = k\Delta_f^{MP}$ where $\Delta_f^{MP} = \frac{1}{T_{cycle}^{MP}} \approx 342.654$ Hz is the frequency spacing derived from the 2D Lissajous sequence length $T_{cycle}^{MP} \approx 2.918$ ms and $k \in \mathbb{N}$ is the frequency index. The index $k$ can be expressed using the mixing factors $m_x$ and $m_z$ as $k_{m_x,m_z} = m_x M_x + m_z M_z$ where $M_x = f_x^{DF} T_{cycle}^{MP} = 76$ and $M_z = f_z^{DF} T_{cycle}^{MP} = 75$, see ref. 47. The figure shows on the left selected frequency patterns (($f$, $m_x$, $m_z$) $\in \{$(51.4 kHz, 2, 0), (102.8 kHz, 4, 0), (155.22 kHz, 3, 3)$\}$) of the $x$-receive chain and on the right of the $y$-receive chain. Each pattern is visualized using an iso-surface rendering (lower left) and three orthogonal slices (upper left: $xz$, upper right: $yz$, lower right: $xy$) where the dotted line indicates the slice position. The complex-valued colormap encodes the amplitude in the saturation and the phase in the color[48]. In the middle part of the figure, the mean SNR over all patches is shown as a function of frequency. To illustrate both, the global and the local SNR progression is plotted for different frequency ranges. The lower part of the figure shows a 3D iso-surface rendering of frequency component 102.80 kHz combined with the actual FFP trajectory derived from the measured fields for the three considered patches. The measured system matrix shows the expected wave-like structure in which the number of extrema depends on the mixing factors[49]. For $m_x = 3$, $m_y = 3$ one can see oscillating

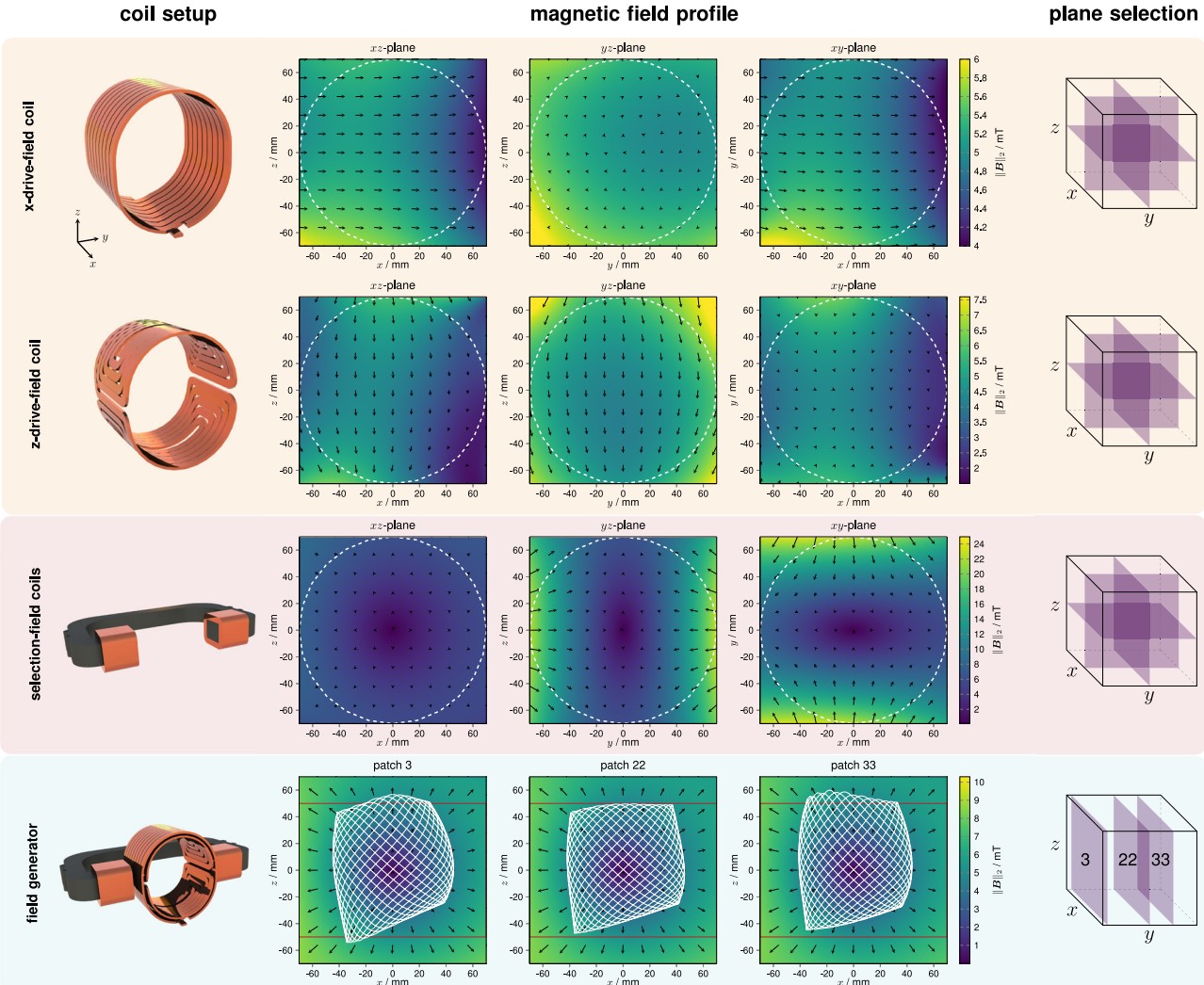

**Fig. 2 | Field analysis of the magnetic field-generating coils.** The field-generating coil setups are shown with the measured corresponding magnetic field profiles and an illustration of the respective planes. The drive field of the *x*- and *z*-channel and the selection field are plotted individually. All fields were measured on a sphere utilizing a three-axis field sensor within the actual setup. For better visualization, the copper shielding between selection-field coils and drive-field coils is not shown. With respect to the entire field generator, the drive-field trajectory of patches 3, 22, and 33 is displayed in the *xz*-plane, while the background features the corresponding selection field shifted in the *y*-direction. For visualization, the frequency ratio is adapted to $\frac{16}{15}$ and the system matrix field of view is outlined in red.

patterns in *x*- and *z*-direction, which shows that the sequence spatially encodes both directions. When considering the patch movement, it is clearly visible that the frequency patterns are shifted when the FFP sweeps slowly in the *y*-direction. However, slight distortions of the patterns are observable, which are caused by field imperfections and violate the true shift-invariance of the system.

Next, we consider the system matrix not as a multi-patch dataset but as a single-patch dataset. This is possible because there are no temporal gaps in the sequence, and thus the sequence can be considered to be 3D with fast FFP movement in the *xz*-plane and slow FFP movement in the *y*-direction. In turn, the sequence time increases to $T_{\text{cycle}}^{\text{SP}} = M_y T_{\text{cycle}}^{\text{MP}} \approx 248.06$ ms and the frequency spacing in turn decreases to 4.031 Hz. This means that the single-patch spectrum contains much more frequencies (factor $M_y$) but that the patch-encoding dimension is lost. This is illustrated in Fig. 4, which now shows in the SNR plot a signal with two levels of sub-bands. The SNR is slightly higher compared to the multi-patch case since an implicit averaging takes place when applying the Fourier transform to the longer time interval. In the single-patch case, the signal occurs at frequencies $f_k^{\text{SP}} = k\Delta_f^{\text{SP}}$ and the frequency index *k* can now be expressed as $k = m_y + M_y(m_x M_x + m_z M_z)$ where $m_y$ is now a new mixing factor that encodes the finest level of frequency shifts. The upper part of Fig. 4 shows selected frequency components for various mixing factors sampling different sub-bands of the frequency space. Again, the expected wave-like patterns are visible, but this time the patterns do not only surround the trajectory plane, but cover the entire FOV and also show oscillating structures in the *y*-direction. The lower part of the figure shows the frequency component 102.7 kHz of the *x*-receive chain in combination with the full 3D FFP trajectory from three different angles. One can again see that both the system matrix pattern and the trajectory are rotated within the *xz*-plane due to field imperfections.

**Sensitivity and spatial resolution experiments.** To determine the sensitivity of the MPI scanner, we implemented the protocol using three different spatial positions developed in ref. 16. First, a dilution series of the tracer *Resotran* was prepared with 50 µL samples inside 200 µL Eppendorf tubes and ascending iron mass between 4 and 512 µg. Applying one full 3D sequence (0.248 s), eight samples were measured at three spatial locations each, positioned along the *y*-axis. This facilitates a distinction between the sample signal and reconstruction artifacts. The reconstruction results are shown in Fig. 5a for several iron masses. The spatial position can be resolved down to 8 µg iron. At 4 µg iron, a blurred dot can be seen, but its position does not change for different sample positions indicating that the seen dot is caused by the system background

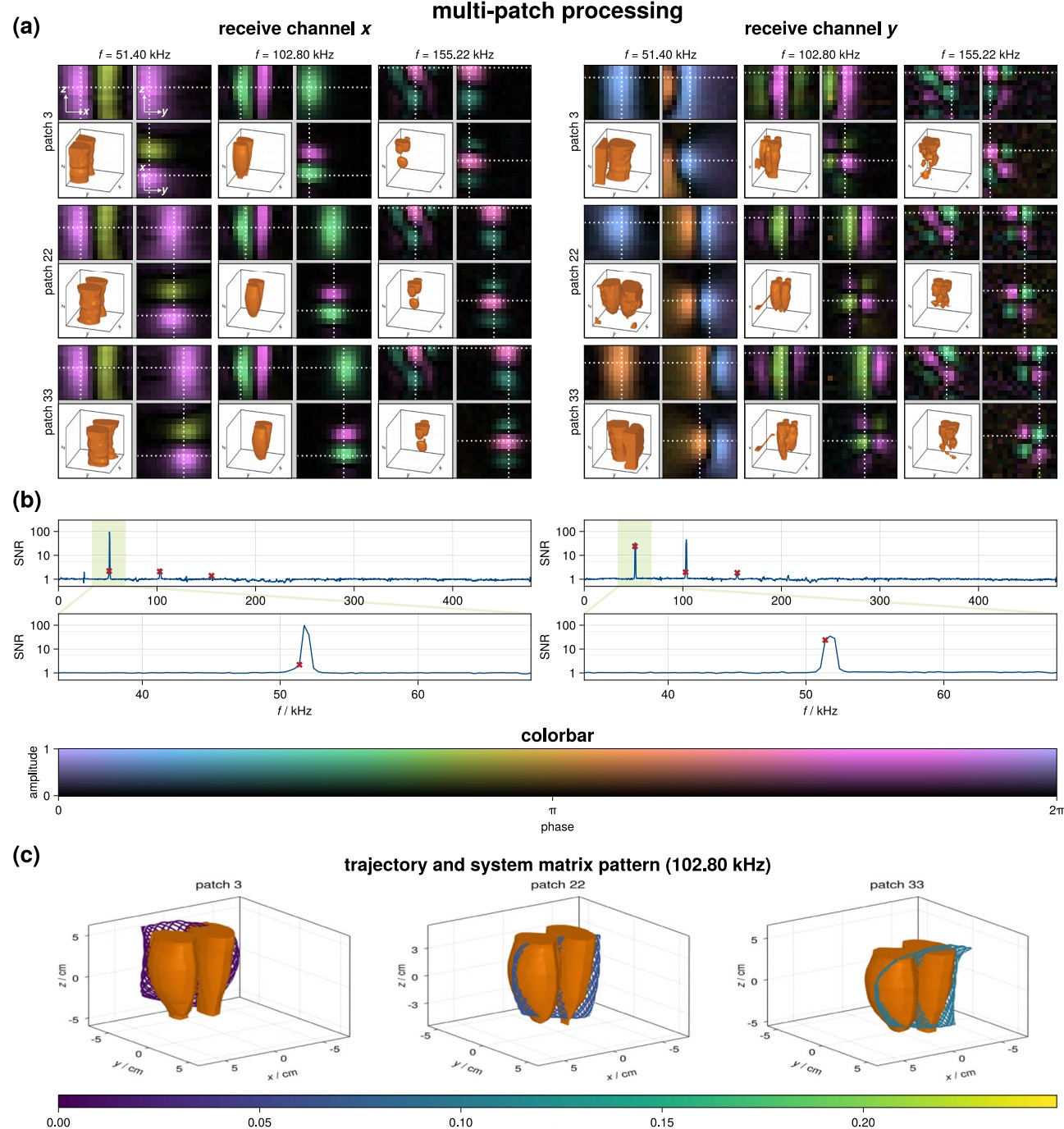

**Fig. 3 | Measured system matrix considering a multi-patch processing. a** Selected frequency components for the *x*-(left) and the *y*-(right) receive channel. Each 3D frequency component is visualized by plotting three orthogonal slices and an iso-surface rendering of the magnitude. The slice position is indicated as a white dotted line, while the frequency component is shown for three different patch positions (*p* ∈ {3, 22, 33}). **b** Signal-to-noise ratio (SNR) of the system matrix rows as a function of frequency. The SNR is visualized in a hierarchical fashion using two different nested frequency ranges, which are indicated by light green boxes. **c** 3D iso-surface rendering in combination with the actual field-free-point trajectory for the considered patch and one selected frequency component. Here, the color encodes the time within the full imaging sequence.

and not by the sample itself. Thus, the detection limit of the scanner is reached at 8 µg iron.

For quantitative analysis, a post-processing step sums the reconstructed particle concentration within the mask around the sample position and multiplies it by the iron mass of the system matrix δ-sample. The results are shown in Fig. 5b. At higher iron mass, the reconstructed iron content matches the applied iron content. For lower iron mass, below 32 µg, the reconstructed particle concentration becomes smaller than expected.

To assess the spatial resolution, a 200 µL δ-sample of *Resotran* with 8.5 mg$_{Fe}$mL$^{-1}$ (152 mol L$^{-1}$) iron was placed in the FOV center. A second, identical δ-sample was mounted on a rod, which could be positioned by a calibration robot. The latter was moved directly next to the centered sample. After imaging using the 3D sequence (0.248 s) without frame averaging, the edge-to-edge distance was increased by 0.5 mm until reconstruction could discriminate the two samples. This procedure was performed for all three main axes. For better reconstruction results, the system matrix grid was

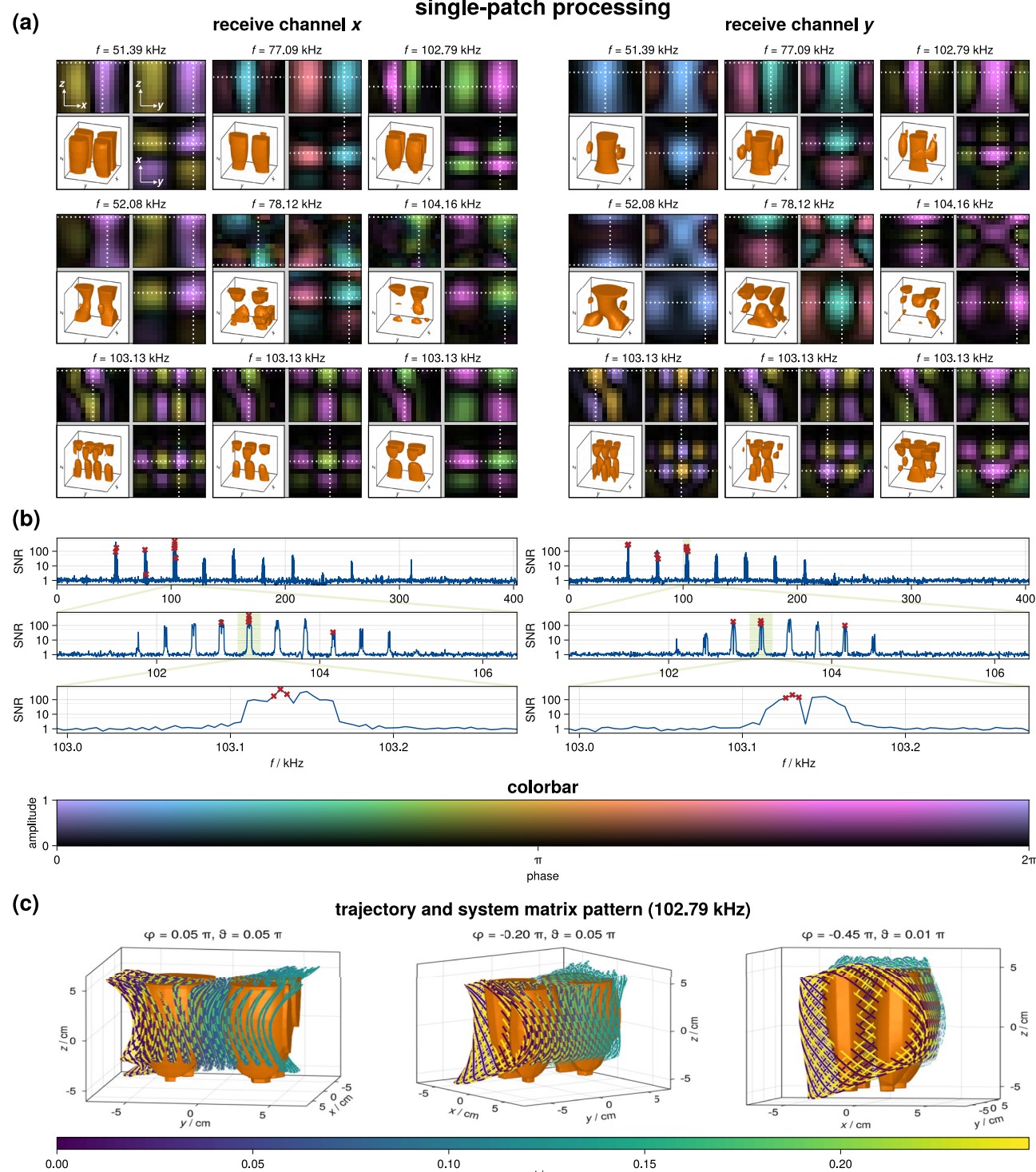

**Fig. 4 | Measured system matrix considering a single-patch processing. a** Selected frequency components for the *x*-(left) and the *y*-(right) receive channel. Each 3D frequency component is visualized by plotting three orthogonal slices and an iso-surface rendering of the magnitude. The slice position is indicated as a white dotted line. **b** Signal-to-noise ratio (SNR) of the system matrix rows as a function of frequency. The SNR is visualized in a hierarchical fashion using three different nested frequency ranges, which are indicated by a light green box. **c** For one selected frequency component, the 3D iso-surface rendering in combination with the actual field-free-point trajectory, viewed from three different angles, is shown. Here, the color encodes the time within the imaging sequence.

interpolated to $25 \times 25 \times 19$. The image signal was summed up over three voxels inside a mask around the direction of interest to generate profile lines. The reconstruction was defined to be resolved if the profile line dropped below half the maximum in the middle between the two samples. In Fig. 5c, the resolved reconstructions are shown for a half and quarter maximum criterion for all the directions. Additionally, the profile lines are shown in the

reconstructed images. With the half maximum definition, the best spatial resolution is found in the *y*-direction with 6.7 mm, followed by 11.2 mm in *x*- and 31.2 mm in *z*-direction.

**Dynamic perfusion experiments.** For the analysis of volumetric imaging at high temporal resolution, we used a flow phantom filled with glass

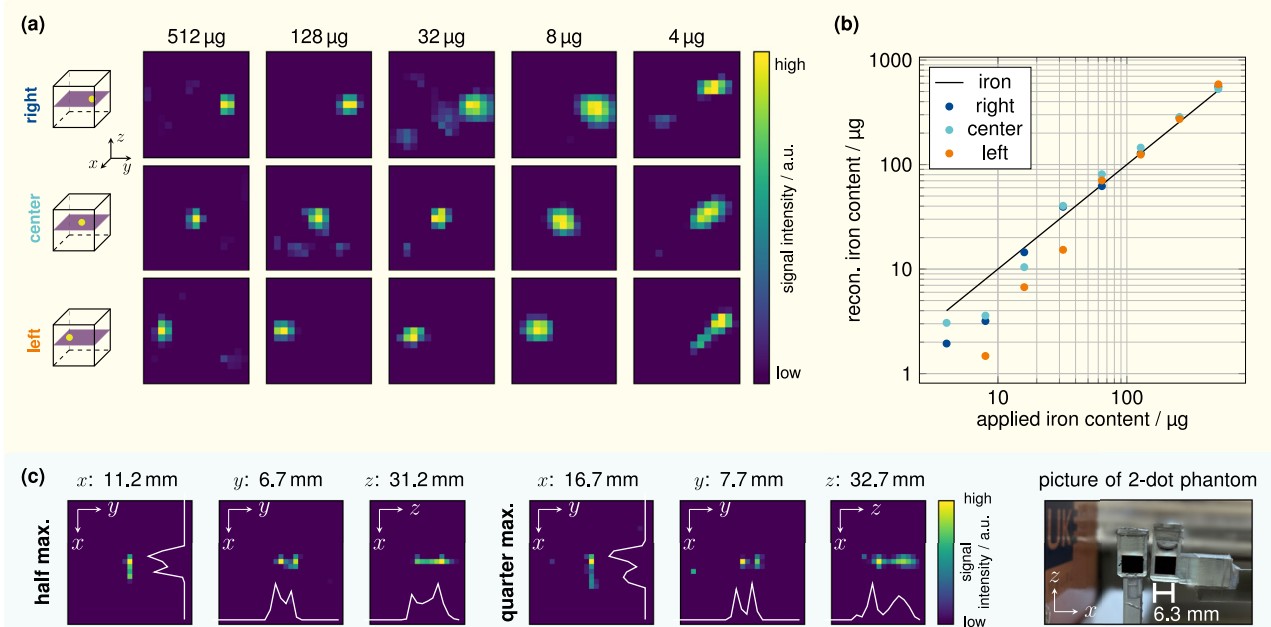

**Fig. 5 | Sensitivity study and spatial resolution experiments. a** Reconstruction results for the sensitivity of descending iron contents inside a 200 μL Eppendorf tube with 50 μL volume of tracer in the central *xy*-plane for multiple positions. **b** For each iron content and each position, the quantitative reconstructed iron content is mapped to the applied iron content. **c** For the spatial resolution experiment using *Resotran*, two δ-samples (200 μL each, 8.5 mg$_{Fe}$mL$^{-1}$ (152 mol L$^{-1}$)) were used. An edge-to-edge distance was achieved by placing one sample in the center of the field of view, while the other was moved by the calibration robot to defined positions with increasing distance (0.5-mm steps). Images were recorded using the 3D sequence with dedicated *x*- and *y*-receive channels, and reconstructions are based on a single frame (one average) with an interpolated grid 25 × 25 × 19. The spatial resolution was considered successful when the profile line in the middle between the two samples dropped below the half maximum (white line). Results for a quarter maximum criterion are shown as well.

spheres and bolus injections of *Resotran*. The phantom in this experiment is similar to the one developed by ref. 8. It consists of two tubes with a volume of 50 mL each, filled with 1 mm diameter glass spheres to simulate capillaries within the tissue. Two rods are placed inside the tubes with evenly distributed holes facing opposing directions, as shown in Fig. 6 on the top right. The tubes represent the two hemispheres of the brain and each tube is connected to a peristaltic pump, which uses suction to deliver an adjustable flow rate that is independent of the other hemisphere. Five experiments were conducted with different average flow rates: While the flow rate in the right hemispheres remained constant at 100 mL min$^{-1}$, 0, 25, 50, 75, and 100% stenosis was obtained for the left hemisphere by reduced flow rates. Prior to experiments, the overall flow rates of outlets A and B were precisely matched by independent reference experiments to compensate for pressure differences. Applying the 3D sequence (0.248 s), each experiment was measured over 150 frames (37.2 s) and a 100 μL bolus of pure *Resotran* (28 mg$_{Fe}$mL$^{-1}$) was administered. For imaging, the same 3D sequence was used for the sensitivity and resolution experiments, which is described in detail in the subsubsection "Imaging sequence". During the perfusion measurements, the observed drive field showed a standard deviation below 0.3% over the 150 consecutive frames without additional control steps. The data processing is divided into a reconstruction step, described in the subsubsection "Image reconstruction", and a post-processing step that is built upon the reconstruction results and yields different perfusion parameters, as described in detail in the subsubsection "Perfusion image calculation". For the reconstruction, the system-matrix grid size was interpolated to $N = n_x \times n_y \times n_z = 25 \times 25 \times 18$ voxels. The reconstruction results revealed that the relevant data for the post-processing step ended after $t = 25$ s.

The results are illustrated in Fig. 6. They indicate that fast dynamic imaging is feasible and different levels of stenosis can be detected and distinguished through calculated perfusion maps. Throughout the referenced figure, a transversal 2D slice is shown, and its relative position is indicated in

the picture of the phantom on the right side. This slice is exemplary for the entire 3D tomogram. The time response graphs in the leftmost column show reconstructed data for voxel A and B, prior to Hann-filtering. Post-processing produced the 3D TTP, MTT, rCBF, and rCBV maps, and the definitions of each perfusion parameter are sketched above each column. In the case of stenosis, the flow suppression is visualized by a delayed signal peak in the TTP map and an increasing transit time in the MTT map (dark colors). Relative blood flow and volume decrease with the severity of the stenosis, which is revealed by lighter colors and a smaller area of outlet A. The decreasing area can be attributed to the threshold mask, which eliminates values below 10% of peak intensity. Slight differences in MTT or rCBV in the case of equal flow rate (0% stenosis) are caused by variations of the phantoms, their filling, and air cavities, which all influence the internal flow. The entire set of four perfusion parameters reveal not only changes due to the stenosis, but also the increments of these changes become visible in each perfusion map.

**Multi-contrast experiments.** To demonstrate the ability of multi-contrast imaging within the MPI brain scanner, a simple two-dot phantom and two different tracers were chosen. The δ-samples of *Resotran* and *synomag* (micromod Partikeltechnologie, Germany) contained 200 μL each with an iron concentration of 8.5 mg$_{Fe}$mL$^{-1}$ (152 mol L$^{-1}$). For this proof-of-concept, applying the 3D sequence (0.248 s), two system matrices were recorded in the *xy*-plane on a 15 × 15 × 1 grid with a system matrix FOV of 140 × 110 mm$^2$ in *x*- and *y*-directions. The δ-samples were used for the imaging experiments, mounted on a 3D-printed platform, and inserted by a calibration robot into the center of the scanner. The imaging sequence was the same as in all other experiments above. A single frame was recorded and reconstructed, using background subtraction. Reconstruction followed the protocol described in the subsubsection "Image reconstruction", with the only difference that two measured system matrices are passed to the Kaczmarz-solver to separate the signal contributions of each tracer sample[31].

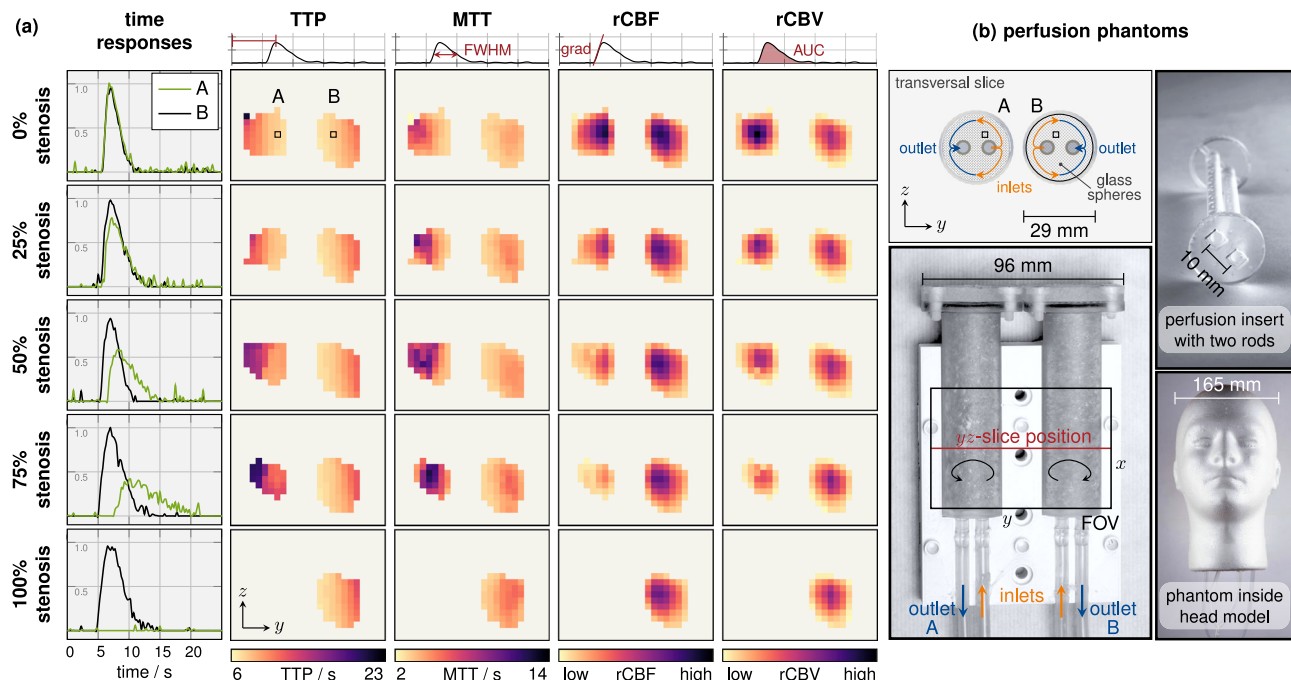

**Fig. 6 | Perfusion experiment for fast dynamic imaging. b** Two cylinders represent the left and right hemispheres of the brain. Each 50 mL cylinder is filled with 1 mm diameter glass spheres. Inside the cylinder is a perfusion insert with two perforated rods, one connected to the inlet (orange) and the other connected to the outlet (blue). They contain multiple holes on opposite sides to mimic tissue perfusion. **a** Perfusion parameters after bolus injection are visualized for the transversal $yz$-slice, that is depicted throughout the reconstructed images. The bolus contained 100 µL of pure *Resotran* (28 mg$_{Fe}$mL$^{-1}$). The measurements were acquired with the 3D sequence. Different levels of stenosis in 25% of steps were simulated by using two independent peristaltic pumps, connected to one output each. The suction was regulated to match a flow rate of 100 mL min$^{-1}$ for the healthy brain half (on the right in each case). For the highlighted voxels A and B, the time responses of the normalized concentration are shown. From the data of the time responses, TTP the time-to-peak, MTT mean-transit-time, rCBF relative cerebral-blood-flow, and rCBV relative cerebral-blood-volume perfusion maps were calculated. These calculations are based on the maximum peak identification, on the full width at half maximum (FWHM), on the gradient (grad), and on the area under the curve (AUC). The rCBF and rCBV were normalized to the maximum value in the imaging volume. All time data were shifted to the arrival time of the bolus.

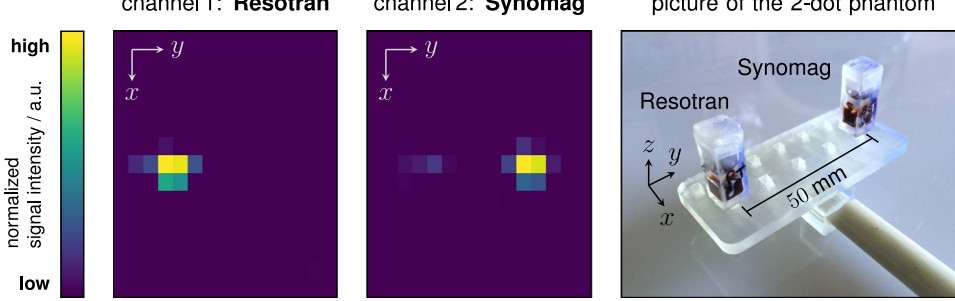

**Fig. 7 | Multi-contrast experiment, using *Resotran* and *synomag*.** The two δ-samples (200 µL each, 8.5 mg$_{Fe}$mL$^{-1}$) were separated by a distance of 50 mm. The image was recorded using the 3D imaging sequence with $x$- and $y$-receive channels, and the reconstruction is based on a single frame (one average). In $x$-direction, the field of view size is 140 mm, while in $y$-direction, it is 110 mm. For each tracer channel (system matrix), the signal intensity is normalized individually. However, the MPI-tailored tracer *synomag* has a stronger particle response. Overall, the separation of the tracer samples was successful.

The resulting concentrations for both reconstruction channels are shown in Fig. 7, along with a picture of the phantom. Both δ-samples are correctly depicted in their respective channel and spatially separated. In the *synomag* channel, a residual signal from the *Resotran* sample can be observed. However, this channel leakage is small compared to the intrinsic signal of the *synomag* sample.

## Discussion

In this study, we investigated a human-sized MPI brain scanner designed to provide 3D imaging with high spatial resolution, 4 Hz temporal resolution, and with a focus on enabling the detection of neurovascular diseases. We achieved a resolution of 12 mm in $x$-, 7 mm in $y$-, and 31 mm in $z$-direction using the clinically approved Fer-ucarbotran tracer *Resotran*. Moreover, our scanner was able to detect an iron dose down to 8 µg. Perfusion experiments were performed and images reveal that five levels of stenosis (25% increments) can be dif-ferentiated. Furthermore, we were able to discriminate *Resotran* and *synomag* within the same FOV in multi-contrast images. Instru-mentation safety was a major focus in the design and implementation of the transmit- and receive chains, as well as obtaining a 3D imaging sequence that samples a 480 mL volume. The scanner can potentially be used in an ICU due to its manageable total power consumption of less than 4 kW and its robustness to electromagnetic interference in an unshielded environment.

## Imaging capabilities

The spatial resolution of an MPI system is an elaborate interaction of gradient-field strength and drive field[50], imaging sequence[51], magnetization response of particles[50], and the receive chain[49]. All measurements were obtained at a DF amplitude of 5 mT in $x$- and 4 mT in $z$-direction, with a gradient strength of $0.24\ Tm^{-1}$ in $y$-direction (measured in the FOV center). Despite being orthogonal to both DF directions, the best spatial resolution of 7 mm was found in the $y$-direction. It benefits from a stronger gradient strength, the dedicated receive coil, and the multi-patch excitation sequence. The $x$-direction benefits from being aligned with one of the DF directions and a dedicated receive coil, but we observe only about half the gradient strength. This culminates in a spatial resolution of 12 mm. Due to the lack of a dedicated receive coil and the same gradient strength as in $x$, the spatial resolution in the $z$-direction is inferior at 33 mm, although it aligns with a DF direction. In terms of sensitivity, a dilution series measured at different positions was required. In contrast to other imaging modalities, noise in MPI can lead to structured image artifacts. For low iron amounts, superimposed noise in combination with strong regularization parameters for the reconstruction algorithm, structured image artifacts structured image artifacts may occur[16].

Compared to the previous version of the head scanner[8], we reached a detection limit of $8\ \mu g_{Fe}$, whereas the old limit was $2\ \mu g_{Fe}$. However, this does not imply that the redesign has a lower sensitivity by a factor of 4, since a direct comparison does not prove to be feasible due to the differences in the experimental configuration and particle system. The reasons for an expected reduction of the sensitivity and spatial resolution are the topology of the imaging sequence that samples a 3D volume in the current version instead of a 2D slice. On the other hand, our new imaging sequence is only half as long, and the signal intensity of the tracer *Resotran* is lower than *perimag* (micromod Partikeltechnologie, Germany), which implies that the actual sensitivity difference is lower.

The dynamic perfusion experiment takes full advantage of the temporal resolution of 4 Hz and its 3D imaging capability. Setting realistic flow rates of $100\ mL\ min^{-1}$, we were able to visualize a bolus passing through our perfusion phantom[52]. In terms of temporal resolution, we calculated four different perfusion parameters in five different levels of stenosis. The spatial resolution of our scanner is sufficient to separate the left and right brain hemispheres in all experimental settings. However, the resolution required to image the cerebral vasculature (brain angiography) is not yet achieved. In a clinical scenario, the size of this bolus would need to be increased, due to the amount of blood delivered to the brain from an intravenous bolus (fraction of 15 to 20%[53]). Assuming a fraction of 20%, the amount of administered boli would be restricted to about 3 per prefilled vial (1.4 mL), although limitations depend on body mass and metabolism[54,55]. A plausible long-term monitoring scenario would require a larger number of administrable boli, to facilitate interventions and evaluate treatment success afterwards. To increase the number of boli, we could reduce the bolus concentration, as the ability of perfusion parameter calculation is likely possible for lower signal strengths of the time response curves. A specifically tailored MPI tracer, which could be clinically approved in the future, would increase the measurement signal leading to a lower iron dose per bolus. A complementary approach could make use of negative boli to increase the total number of administrable boli[56].

Multi-contrast imaging was performed utilizing the clinically approved tracer *Resotran* and the MPI-tailored tracer *synomag*. For this proof-of-concept, we used 200 μL samples at a concentration of $8.5\ mg_{Fe}mL^{-1}$ ($152\ mol\ L^{-1}$) and a single measurement without averaging. We observe minor channel leakage in the multi-contrast tomogram, which is typical for multi-contrast imaging[31]. In general, the underlying phenomena for the observed channel leakage in multi-contrast imaging needs to be further investigated. However, localization and discrimination of both samples was successful and has the potential to provide additional information like temperature in hyperthermia[29,57].

All experiments in this work have been performed at least twice to ensure reproducibility. Qualitatively, no significant difference could be found. The system-characterizing values obtained from the experiments are given without an accuracy estimate, but provide a good estimate of the corresponding order of magnitude. Future applications can use the high frame rate of the scanner to aggregate and statistically evaluate the acquired data.

Reconstructions of the presented results were performed retrospectively on the stored measurements. However, the operational control and data processing software makes it possible to reconstruct online. Although the software is not yet optimized for this use case, the GUI is able to perform online reconstruction with a latency of less than one second between signal acquisition and image visualization. Further latency reduction could be achieved by accessing the measurement data earlier in the processing pipeline and by adapting additional methods from ref. 58.

Common to the imaging sequence of the current and old scanner version, is the combination of a slow selection-field shift in $y$-direction with fast orthogonal DF excitation. The old scanner used a 1D drive field in the $x$-direction, which results in an imaging trajectory that samples a 2D FOV in the $xy$-plane at 2 Hz. In contrast, the current design uses a Lissajous-type DF excitation in $xz$-plane, sampling a 3D FOV (480 mL; the volume of an adult human brain is about 1200 mL[59]) at 4 Hz. We note that both sampling trajectories are redundant by a factor of two since they contain a sweep along the positive and negative $y$-direction. This yields the potential to increase the frame rate by a factor of two when reconstructing the two halves independently, as has been proposed by ref. 60. A major improvement of the redesign is the extension of the FOV to a 3D volume, sampled at twice the frame rate, which is a leap towards imaging the entire human brain. Prototypes of multi-coil iron core selection-field generators are in development, that further enlarge the FOV and extend the space of feasible imaging sequences[34,61].

## Device safety

The presented scanner in this work uses the same selection-field yoke as the prior version by ref. 8, however, all other major hardware components (the DFG, ICN, HCR, transmit, and receive chain) were redesigned for this scanner upgrade to achieve 3D imaging and focus on human safety. To this end, the limitation of high voltages in the vicinity of the imaging volume and the implementation of an independent surveillance unit were key concerns.

On a path toward human trials, safety concerns and regulations regarding the specific absorption rate (SAR), peripheral nerve stimulation (PNS), and conductors in patient proximity influence the scanner design, and imaging sequence and limit maximum magnetic field strengths. The SAR limitations in the head are given with $3.2\ W\ kg^{-1}$ for the chosen DF frequencies in the kHz range[57,62]. However, for sinusoidal electric fields below 100 kHz, PNS concerns prevail and are the limiting factor for alternating field strengths[63–65]. Heeding PNS limitations, the maximum DF amplitude was set to 5 mT, which is realistic for human trials[66,67]. Compared to MRI gradients, the maximum slew rate of the dynamic selection field ($\approx 24\ Tm^{-1}s^{-1}$) is very low and below the risk threshold for PNS[68,69]. In MRI, all ferromagnetic components must be excluded from entering the scanner, including pacemakers, mechanical ventilators and oxygen cylinders[1]. Potentially, this does not apply to our scanner, due to the confinement of high magnetic fields within the head region only. Further investigations on specific device compatibility are required.

Electrical safety regarding shock and discharge, are mainly addressed by reducing the inductance of the DFG. The necessary power to obtain the same field strength is thus provided by a high coil current, which creates a maximum voltage of 535 V (240 A at 5 mT, 14.4 μH) that is a reduction of a factor of about 4 compared to the previous version of the DFG[8]. To achieve the capability to conduct around 300 A with Litz wire, we used Rutherford wire parallelization in both DF coils, which also minimizes the proximity effect. Moreover, the polyamide housing of the DFG provides dielectric isolation and increases breakthrough voltage between DFG components and the patient. Due to the ICN, the entire HCR obtains a floating potential, which implies that touching a single exposed point of the circuit is not a hazard, because residual leakage currents towards the ground, e.g., caused by

capacitive coupling, are below 20 mA[70]. Thus, the insulation would have to fail at two separate points simultaneously, and both would have to make contact with the patient to create a dangerous voltage drop across the body. The operational control of the scanner is implemented with multiple safety mechanisms, features an independent monitoring of relevant metrics, and has the ability to intervene in signal generation. For example, the SU is capable of disabling the transmission chain in the event of overheating. The same applies to the resonant tuning heating unit. An active imaging phase is only entered if no unexpected DF feedback is observed. Unintended high DF levels due to incorrect inputs or component failures are prevented by the chosen operating point at the upper power limit of the amplifier. In addition, the resonant transmit chain is frequency-specific and unexpected waveform changes or detuning of components in the transmit chain will reduce the power in the DFG. Finally, a human operator can use a hardware console to disable parts of the system at any point.

It should be noted that only phantoms were used in the studies presented. However, the corresponding measurement parameters were carefully chosen to be reasonable for future studies in humans. Ultimately, conclusive assertions concerning safety and tolerability in humans can only be made after animal or human studies[71].

## Hardware implementation
The design concept for signal generation and reception, includes a symmetric approach to enable simultaneous signal generation and reception[72], called send-receive approach (TxRx). This utilizes a pick-up node within the HCR that is sensitive to the voltage signal induced by the particle magnetization response, while suppressing the excitation voltage. The implementation of this approach requires an ICN as one fundamental part. If built symmetrically with a similar serial resistance of the inductor to the DFG, twice the power is required to match the field generated by a single resonance circuit, limiting the maximum DF amplitude of the TxRx approach. In addition, the complexity of the send-receive chain increases, leading to a high susceptibility to disturbances generated within the HCR (e.g., from connections and eddy currents). In our case, the received signal acquired by this approach was inferior to the received signal of a dedicated receive coil, which is able to suppress signal distortions[73], justifying the utilization of the dedicated x-receive coil for this channel. A dedicated receive coil in the z-direction has not yet been developed due to the laborious design and intricate tuning process required for the cancellation coil. Due to the DF saddle coil, the magnetic field profiles in z have only a small homogeneous area, and coupling to other channels becomes an issue.

The sampling trajectory originates from a superposition of the slowly varying selection field and the fast-oscillating drive field. The arising FFP movement follows a trajectory, which samples the FOV and ideal MPI systems try to generate homogeneous and orthogonal drive fields with a linear selection field. However, to achieve low field imperfections, large coils are required that are less energy efficient. Restrictions of design space, power supply, and coil coupling[74] within the DFG of our scanner cause noticeable field imperfections, as shown in Fig. 2, that deviate from the ideal field. Especially towards the edges of the coils, field imperfections are severe and cause a deformation of the Lissajous-type trajectory in all spatial dimensions. Our magnetic field measurements contain systematic uncertainties due to errors in coil sensor size and orientation. Consequently, they can only shear and rotate the resulting trajectory, and the observed deformations can be attributed mainly to the imperfections of the magnetic fields.

## Conclusion and outlook
We have performed a comprehensive system characterization of a 3D human-sized MPI scanner for real-time cerebral applications. The perfusion experiments conducted provide a proof-of concept that the discrimination of brain hemispheres and different severities of stenosis are possible. This will allow us to better assess which clinical application scenarios are feasible in the future. For example, MPI trials for the ischemic stroke scenario in human volunteers may soon be possible using the *Resotran* tracer. Tailored MPI tracers, such as *perimag* and *synomag*, with future clinical approvals, promise to increase system performance in sensitivity and spatial resolution and further expand the range of applications.

## Methods
### Field generation and reception
This section provides a detailed presentation of all scanner components and introduces the concept of how they interact to obtain MPI measurements. Beginning with the generation of magnetic fields, details are given on signal reception, imaging sequences, operation control, and data processing for image reconstruction and post-processing.

The scanners fundamental component for signal generation and acquisition is realized by a stack of three RedPitaya STEMlab 125-14 (RPs). These are a flexible low-cost hardware solution for integrating DACs and ADCs into a single device for precise real-time signal handling. The open-source software for the RPs by ref. 36, ensures a parallel and synchronous signal generation and reception by synchronizing the 125 MHz clock and logic of the three used RPs. This stack thus provides six radio-frequency input and output channels.

**Drive-field generation.** In Fig. 8, a simplified equivalent circuit diagram (ECD) of the transmit circuit is shown, with details of the transmit filter for one of the two channels. In the following, we describe the components of the transmit chain in detail from field to source: the drive-field generator (DFG), the high current resonator (HCR), the inductive coupling network (ICN), the transmit filter and the impedance matching transformer at the output of the amplifier.

Drive-field generator. The MPI brain scanner utilizes two DFCs in x- and z-direction, respectively. The solenoid x-DFC (with inductance $L_x$) and the saddle coil z-DFC (with inductance $L_z$) are nested, with the x-DFC placed on the inside. Both are manufactured using Rutherford wire parallelization with 12 individual strands of a high-frequency Litz wire (2000 isolated strands with 50 μm diameter, Elektrisola, Germany). It further serves the minimization of the self-inductance of the wire, keeps individual wires at identical lengths, and mitigates proximity- and skin-effect. By parallelizing 12 Litz wires, a minimized serial resistance is achieved while maintaining the ability to form and wind the wire into the desired coil topology[75]. In the FOV center, the manufactured DFCs exhibit coil sensitivities of 0.022 mTA$^{-1}$ and 0.014 mTA$^{-1}$ in x- and z-direction, respectively. Further details on the achieved component values can be found in Fig. 8. The support structure (PA2200) of the DFG forms an elliptic open bore with a width of 17.5 cm and a height of 21.5 cm. The resulting 2D drive field is set up in the xz-plane with DF frequencies set to $f_x = \frac{125}{4864}$ MHz ≈ 25.699 kHz and $f_z = \frac{125}{4800}$ MHz ≈ 26.042 kHz. The frequency ratio between the two channels results in a closed 2D Lissajous trajectory after 76 and 75 periods for the x- and z-directions, respectively.

High-current resonator. In order to mitigate high voltages near the human brain, a design objective was to utilize parallelized low inductance DFCs that require high currents instead of high voltages for generating the desired drive field. To minimize reactive power and obtain symmetry, the DFCs are operated in resonance at the DF frequency by connecting two capacitors of equal size both upstream and downstream of the inductance. Instead of utilizing a capacitive voltage divider for impedance matching[76], a mirrored resonant setup with a toroidal transformer coil ($L_{ICN2}$) is connected to each DFC with the same resonance frequency. For resonance tuning, the capacities $C_{Hx_i}$, $C_{Hz_i}$, $i = 1, ..., 4$ are temperature controlled capacitors (CSM 150/200, Celem, Israel), which enables stable and precise resonance tuning[77].

Manufacturing orthogonal DFCs is challenging; hence residual coil coupling between DF channels must be addressed in order to avoid beat frequencies, undesired frequency shifts by mode splitting, and resulting losses. Due to the resonant behavior of the coupled coil circuit and the low difference of DF frequencies, the coupled signals also experience an amplification in the other circuit. Even the small coupling coefficient of $k_c = 0.06$ can lead to large currents in the orthogonal coil, resulting in a

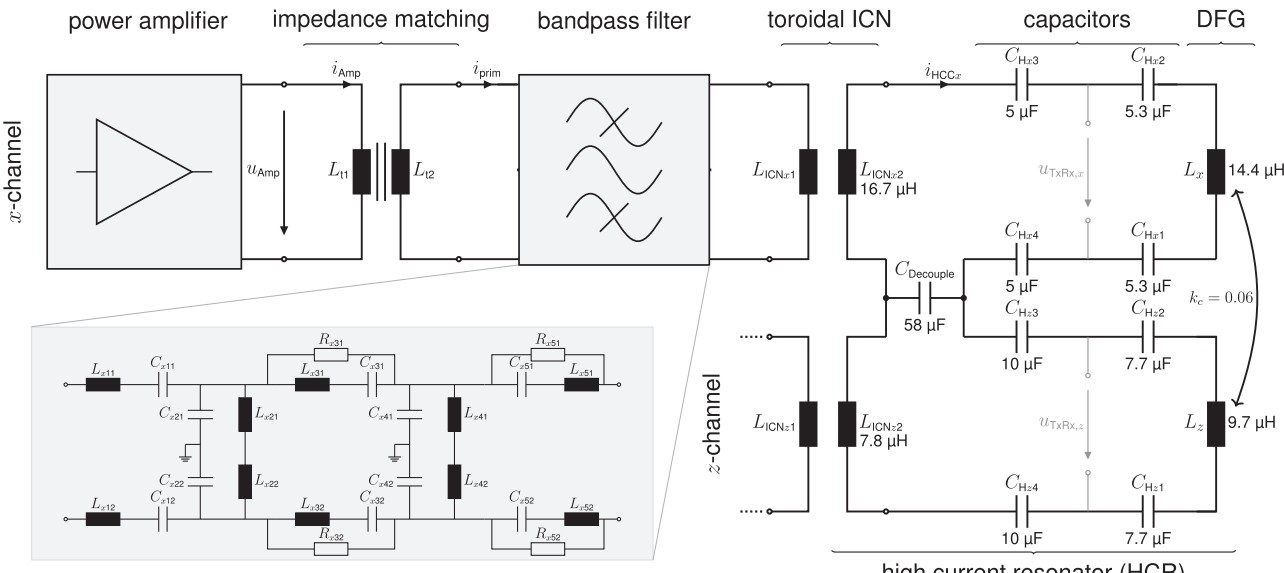

**Fig. 8 | Simplified transmit circuit.** The $x$- and $z$-drive-field (DF) excitation channels are designed to provide resonances at 25.699 and 26.042 kHz, respectively. Both channels consist of their own power amplifier, impedance matching transformer (with an iron core, ‖) and band-pass filter. The band-pass filter is shown for $x$ only, but is identical for the $z$-channel. The inductive coupling networks (ICNs) were individually developed with different litz-wire parallelizations (higher current in $z$)

and roughly matched the inductance of the corresponding drive-field coils within the drive-field generator (DFG). Resonance is achieved by the high current resonator (HCR) circuit. Residual coupling of the DF channels and decoupling countermeasures are also shown. Values of major components are denoted in the equivalent circuit diagram. Since fine-tuning needs to be performed with a fully assembled HCR, the denoted HCR capacity values are guide values.

strong beating signal of the two frequencies. For the HCR, the field coupling far exceeds 10%, causing a distorted trajectory, which substantially degrades the image quality[74]. Coupling is reduced by a decoupling capacitor ($C_{Decouple}$) that is connected in series within both circuits. It carries both currents and matches the complex conjugate impedance of the mutual inductance $L_M = k_c \sqrt{L_x L_z}$ between the two DFCs. As a result, the voltage drop across the capacitor cancels the induced voltage from the other channel within each channel loop. For our setup, we built a capacitor $C_{Decouple} \approx 58\,\mu F$ by parallelization of $58 \cdot 1\,\mu F$ (MKP C4Q, Kemet, USA) and achieved a residual channel coupling of $-35$ dB.

Inductive coupling network. The aforementioned toroidal air-core transformer coil composes the inductive coupling network (ICN), one for each DF channel. The ICN serves the triple purpose of high current gain (impedance matching), circuit symmetry, and floating potentials. The last point is achieved by any transformer. It increases patient safety by requiring direct contact with two separate points of the circuit in order to cause a voltage drop across a grounded person. Also, a floating HCR avoids ground loops that may have a negative influence on signal reception.

The intention of circuit symmetry is to obtain two voltage nodes in each channel, between which the voltage of the fundamental frequency is zero. Both inductors of a channel (ICN and DFG) are in a series resonance with their corresponding capacitors, which have equal voltages but opposing sign at resonance. A pick-up point for the particle harmonics is thus created, which is nested in between the capacitors within the HCR, denoted by $u_{TxRx}$ in Fig. 8, with a suppressed feedthrough[72]. For the higher harmonics induced by the particles, the inductors on both sides result in higher impedances, creating an inductive voltage divider for both noise and particle signals.

The ICN also plays a crucial role in providing current gain at the resonance of the HCR. This enables the transformation of filtered transmit signals into the desired low-voltage-high-current signal. Multiple Litz-wire parallelization on the secondary transformer side ($L_{ICN2}$) are used to achieve low losses and a high-quality factor; nevertheless, additional inductance and resistance is introduced by the ICN into the HCR. The HCR constitutes the load of the transformer, and the primary transformer coil ($L_{ICN1}$) shows a real load impedance ($Z_{ICN} \approx 30\,\Omega$) at the resonance frequency and becomes

part of the last stage of the transmit filter. The achieved current gain is 31 for the $x$-channel and 35 for the $z$-channel.

Transmit filter. A differential 5th-order band-pass filter is used to smoothen the excitation voltage before it is connected to the primary side of the ICN, shown in Fig. 8. The three odd stages of the filter consist of serial resonance circuits that act as a band-pass by having a minimum impedance at resonance. In-between these stages, two even stages act as parallel resonators that create a short for all other frequencies in order to dissipate the energy of undesired frequencies. The serial resonator (odd stages) is composed of a 3D-printed toroidal air coil ($L_{s,odd} \approx 500\,\mu H$, $R_{s,odd} \approx 1\,\Omega$). For a symmetrical differential signal, the toroid is separated into two halves, with each half forming their own resonance together with a high voltage, polypropylene film capacitor (e.g., KEMET C4C series, 2 kV DC rated). Due to the mutual magnetic field within the toroidal coil, the field lines run through both sides of the toroid. Additional resistors across the second and third serial resonator stages are utilized to attenuate side lopes[76]. The two parallel resonators (even stages) are formed by smaller toroidal air coils ($L_{s,even} \approx 250\,\mu H$, $R_{s,even} \approx 500\,m\Omega$) to pose a high impedance at the DF frequency between the two voltage rails. The assembled transmit-filter chains attenuate harmonic distortions of the DF signal by $-65$, $-100$, and $-150$ dB amplitude ratio for the second, third, and fourth harmonics, respectively. The overall differential filter setup provides common mode rejection.

Transformer. For maximum power transfer, we implemented an impedance matching transformer with an iron core (N87 material B65686A0000R087, TDK Electronics, Germany). To prevent distortion and harmonics due to core saturation effects, the core flux density is minimized to 16% of the saturation magnetization $B_{sat}$[76]. The secondary side of the transformer is connected to the transmit filter and ensures floating potentials and differential signaling. The turns ratio of the transformer changes the low impedance ($\approx 2.5\,\Omega$, amplifier side) to a high impedance ($\approx 30\,\Omega$, filter side) to minimize the current in the transmit filter.

Drive-field power amplifier. Two 1200 W power amplifier (A1110-40-QE-100, Dr. Hubert GmbH, Germany) are used in voltage mode for

amplification of each DF signal. To generate the DF strength of 5 and 4 mT, a total power of 930 W for $x$ and 1100 W for $z$ has to be provided by the power amplifiers. For safety and control reasons, the initial signal from the DAC runs through a relay at the input of the amplifier that is only closed during measurements (by the SU). In addition, serial interlock commands are sent to the amplifier to ensure that the output is only enabled during a measurement. The advantage of this double safety configuration is that the SU is able to interrupt transmission in case of failure (e.g., temperature overshoot), even when the software activated the amplifier via the interlock mechanism. A pre-amplifier is used to amplify the voltage signal by a factor of 8 to scale it to the required input voltage of the power amplifier.

Feedback signal. To control the DF signal in amplitude and phase, one turn is wound around each ICN toroid. According to the law of induction, the induced voltage is proportional to the field, which is proportional to the HCR current and hence to the drive field. The induced signal is fed back to the RPs via a voltage divider, where it is processed during the DF control phase. To generate a stable DF trajectory, the control accuracy is set <1% and able to generate field strengths up to 5.5 mT for the $x$-channel and 4.5 mT for the $z$-channel.

**Selection-field generation**. The previously presented SFG in ref. [8] is used to generate the required gradient field, and further moves the FFP in $y$-direction with a maximum displacement amplitude of 70 mm from the center. The SFG consists of two coils ($L_G = 200$ mH) mounted on a soft-iron yoke at a distance of 31 cm from each other. The magnetic gradient field is generated by superimposing the two field contributions, in a Maxwell-like coil topology with opposing current directions. The setting of equal opposite coil currents creates an FFP in the FOV center. By changing the currents, the position of the FFP along the $y$-axis can be shifted and the maximum FFP offset is set to 43 mm to each side with respect to the FOV center. We chose the FFP velocity to be 68.8 cm s$^{-1}$, which results in a similar trajectory density in the $y$-direction as in the $xz$-plane (see subsubsection "System matrix analysis"). The selection-field signals are generated by the aforementioned RedPitaya STEMlab 125-14 (RPs). A pre-amplifier is used to scale the input voltage according to the required voltage for the power amplifier for selection-field generation. For this purpose, two AE Techron 2105 (AE Techron, USA) in current mode feed the selection-field coils. For the generation of the gradient, a total power of 380 W is required.

## Signal reception

Receive coils. Two dedicated receive coils are used for signal reception for all measured data in this work. The gradiometric $x$-coil has ten turns on a length of 10 cm and 18 counter turns at the rear end of the DFG. In the $y$-direction, a saddle coil with two times 20 turns is installed. Due to the orthogonal orientation of the two DFCs, a gradiometric setup is not required, and residual feedthrough voltages are suppressed by the receive filter. Both dedicated receive coils are located within the 3D-printed housing of the DFG, on the inside of the DFCs. A dedicated $z$-receive coil has not yet been implemented, due to its challenging design as a gradiometric saddle coil. The small homogeneous field region of the $z$-channel does not allow a clear spatial separation of the FOV and the sensitive region of the gradiometer.

However, as mentioned in the subsubsection "Drive-field generation" in the ICN paragraph, the symmetric circuit design included a second method for signal pick-up, referred to as the send-receive method (TxRx). Consequently, receive signals for $x$ and $z$ can be obtained in principle. The pick-up node is positioned in the LC circuits of the HCR between $C_{H2}$, $C_{H3}$, and $C_{H1}$, $C_{H4}$, for each DF channel (see Fig. 8). In theory, the voltage of the fundamental DF frequency is near zero at this node, and any induced particle signal will create a voltage drop across the DFC (and the parallel ICN, halving the received signal if $L_{ICNx2} \approx L_x$, or $L_{ICNz2} \approx L_z$). Between these nodes, the input of the receive chain can be connected, enabling signal reception during transmission[72].

Receive chain. The receive voltage $u_{Rx}$ is composed of the particle signal, the background noise, distortions, and the direct signal feedthrough from the DFCs. Before the signal is connected to the LNA, a fourth-order resonance notch filter suppresses the fundamental frequencies around 26 kHz (20 to 33 kHz stopband). The filter is a differential circuit, shown in Fig. 9a. The odd stages feature a parallel resonance with high impedance at the DF frequency, and the even stages are serial resonances towards the ground with low impedance at the DF frequency. To avoid nonlinear effects due to high receive voltages, the first two stages utilize toroidal air coil resonators. The attenuated residual voltage in the last two stages allows the use of ferrite iron core coils (B64290, TDK Electronics, Germany). The filtered receive signal is connected to an improved version of the custom LNA by ref. [8] via a signal-matching transformer. The transformer and the input impedance of the LNA are used to shift the resonance frequency of LNA and receive coil to optimize signal amplification of higher harmonics in the 300 to 600 kHz range[78]. The LNA consists of three amplification stages, the first stage is built

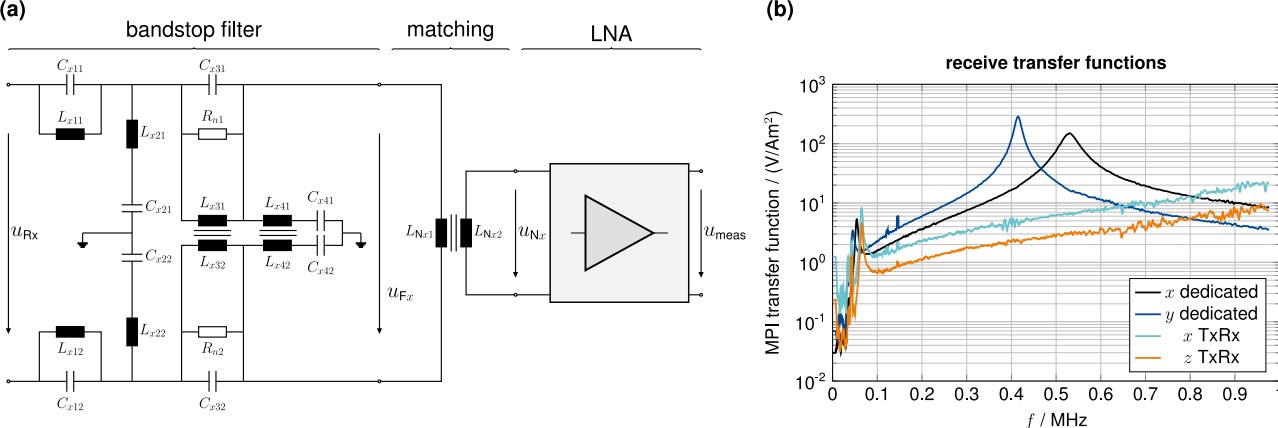

**Fig. 9 | Receive path. a** Equivalent circuit diagram of a receive filter circuit with signal matching for the low noise amplifier (LNA). In the receive circuit, a four-stage analog band-stop filter is installed for $x$-, $y$-, and $z$-feedthrough suppression, although only $x$ and $y$ are used for reconstruction in this work. The first two filter stages are utilized by air coil resonators and the last two stages by iron core coil resonators. A signal-matching transformer follows the filter to adjust the received

signal prior to the LNA. For differential signaling, the LNA output is connected to a differential amplifier, and in front of the analog-to-digital converter, an instrumentation amplifier generates the single-ended measurement signal $u_{meas}$. **b** The receive path Magnetic Particle Imaging (MPI) transfer function for the transformation of $u_{meas}$ into the magnetic moment $m_{meas}$ is plotted for the two dedicated receive coils and the two signal paths using the send-receive approach (TxRx).

by parallelizing 20 JFETs (BF862, NXP Semiconductors, Netherlands), followed by two stages with non-inverting operational amplifier circuit using dual ICs (ADA4898, Analog Devices, USA)[79]. The output is converted into a differential signal by a fully differential amplifier (AD8138, Analog Devices, USA). Just before the input of the ADC of the RPs, an instrumentation amplifier (AD8253, Analog Devices, USA) generates the single-ended measurement signal $u_{meas}$.

For each of the four receive chains, the receive path calibration was performed utilizing a custom calibration coil[80]. In Fig. 9b, the individually measured MPI transfer functions are plotted. The two dedicated receive coils provide resonances at 531 and 414 kHz for $x$ and $y$, respectively. It can be seen that the TxRx signal for $x$ and $z$ are less sensitive in the relevant 100 to 800 kHz range.

**Imaging sequence**. The MPI brain scanner superimposes two fast drive fields in the $x$- and $z$-direction and a dynamic selection field. The selection field is shifted along the $y$-axis, as explained in previous sections. Using and combining these constituents, it is possible to perform various imaging sequences. First of all, there are three different 2D sequences possible that drive the FFP either along a 2D Lissajous trajectory with a frequency ratio of $\frac{75}{76}$ ($xz$-plane) or along Cartesian trajectories ($xy$- and $yz$-plane) with flexible densities. For an overview of common MPI sampling trajectories, we refer the reader to ref. 51. In the case of the $xz$-sequence, both drive fields are activated, and a static FFP field is generated at an eligible $y$-position, using the SFG. For the other two cases, only one DF channel is activated, and the FFP is swept dynamically along the $y$-direction. The repetition time of the $xz$-sequence is fixed and given by $T_{cycle}^{xz} = 2.918$ ms and for the other two sequences, $T_{cycle}^{xy} = T_{cycle}^{yz} = 0.25$ s if we set the $y$-sequence to 4 Hz (the default value for the scanner). We note, that the first generation of the scanner published in ref. 8 was 2D only and limited to the $xy$-sequence with a frame rate of 2 Hz.

In addition to the 2D sequences, it is also possible to apply fully 3D sequences. This is done by using both DF channels and simultaneously generating a slowly varying selection field with an FFP moving along the $y$-axis[81]. The resulting sampling trajectory is a 2D Lissajous trajectory within the $xz$-plane that is slowly swept back and forth along the $y$-direction, as visualized in Fig. 1. In this way, the volume is sampled plane by plane, and the repetition time is derived from the $y$-sequence, resulting in $T_{cycle}^{xyz} = 0.25$ s for a 4 Hz sequence. The waveform of the currents applied to the selection field is chosen to be triangular, such that the sequence is always periodic, at constant velocity, and without sudden discontinuities, in order to avoid strong mechanical forces in the copper shield of the DFG due to eddy currents. Furthermore, a triangular sweep minimizes the likelihood of PNS and SAR by the selection field. The triangular waveform provides a constant slew rate, and results in an almost linear FFP movement as the iron cores are not yet saturated. The chosen sequence type depends on the spatial dimensions and the requirements for temporal resolution. Within this work, we only show experiments that were performed using the 3D imaging sequence and therefore focus on this particular sequence in the following. The nominal FOV captured by the 3D sequence assuming ideal magnetic fields is $(84 \times 85 \times 67 \text{ mm}^3)$. The density of the FFP sampling trajectory within the $xz$-plane is inhomogeneous due to the sinusoidal excitation[51], and the largest gap can be derived from the Lissajous node points[82]. For the applied Lissajous sequence, the resulting distance is 1.819 mm. The distance between slices in the $y$-direction is equidistant with 1.982 mm. Both values are well below the expected spatial resolution between 5 and 20 mm, which ensures that the resolution is not limited by the sampling scheme.

**Operational control**. The task of the operation control is to implement the chosen imaging sequence by coordinating all scanner components used during measurement. Next to the synchronous signal handling, an imaging sequence also requires several asynchronous tasks, such as enabling the various amplifiers, moving a calibration robot, updating the

RPs waveforms based on different imaging sequences, or informing the SU of an upcoming signal generation. These steps are handled by the open-source framework[37], which can implement a variety of different measurement scenarios, including system matrix calibrations and magnetic field measurements[40].

Depending on the type of measurement and sequence, the components and steps required may vary, but in general, each measurement can be divided into three phases. During the setup phase, the amplifiers are disabled, the RPs are neither transmitting nor receiving signals, and the resonance tuning heating is enabled. In this phase, various components are prepared for the next imaging sequence, e.g., the calibration robot is moved, or the RPs waveforms are updated. The next phase is the control phase. This is where the DF amplifiers are enabled, and the DF waveform is sent down the transmit chain. During these transmissions, the feedback signal is analyzed at the frequencies of the DF channels, and the transmitted amplitude and phase are adjusted to the desired values from the selected sequence. When the deviation between the selected and observed DF waveform is sufficiently small, the control phase ends. In measurement scenarios with frequent measurements, such as a system matrix calibration, the feedback signal from the previous measurement can be used instead to adjust the amplitude and phase. The third phase is the active imaging phase. Here the resonance tuning heating is switched off to prevent possible distortions, both the SF and DF waveforms are transmitted, and the signal is received via the RPs to be stored for further processing.

**Data processing**

**Image reconstruction**. The measured voltage signal of the $j$-th time frame $u_j(t, l)$ depends on the time $t \in \mathbb{R}_+$ and the receive channel $l \in \{1, \ldots, L\}$ where $L$ is the number of receive channels, i.e., $L = 2$ in our case. The signal is sampled at time points $t_i = (i - 1)/f_{sample}$, $i = 1, \ldots, K$ where $f_{sample} = \frac{125}{64}$ MHz and $K = 484,500$ for our 3D 4 Hz (0.248 s) sequence. Prior to reconstruction, we apply the standard signal processing[9], which involves Fourier transformation, a frequency selection based on a band-stop filter in the receive chain filter (see above for details), as well as a filtering based on the SNR using different thresholds ranging from 3 to 100, resulting in $M$ frequency components. For better readability of the following paragraphs, we express the resulting discrete signal as the vector $\mathbf{u} \in \mathbb{C}^M$, omitting the frame index $j$.

The goal of image reconstruction is to recover an image $\mathbf{c} \in \mathbb{R}_+^N$ that is discretized on a (3D) grid with $N$ voxels. The relationship between $\mathbf{u}$ and $\mathbf{c}$ is given by the linear system of equations $\mathbf{Sc} = \mathbf{u}$, where $\mathbf{S} \in \mathbb{C}^{M \times N}$ is the system matrix that encodes the physical process from the particle magnetization progression to the system-dependent measurement signal at the end of the receive chain. Since the linear system is ill-conditioned and the measured data is disturbed by noise, we consider a regularized least-squares approach

$$\underset{\mathbf{c} \in \mathbb{R}_+^N}{\text{argmin}} \parallel \mathbf{Sc} - \mathbf{u} \parallel_2^2 + \lambda_{L^2} \parallel \mathbf{c} \parallel_2^2 + \lambda_{L^1} \parallel \mathbf{c} \parallel_1, \quad (1)$$

where the first term ensures data consistency, the second term penalizes large solutions and prevents that the particle concentration vector $\mathbf{c}$ is fitted to the noise in the measurement $\mathbf{u}$, and the last term allows to penalize non-sparse solutions, which helps in reducing noise. The optimization problem (1) is solved using the iterative Kaczmarz approach[83] using $L^2$-regularization as well as $L^1$-regularization[84,85]. The reconstruction has four parameters in total: $\lambda_{L^2}, \lambda_{L^1}$, the SNR threshold, and the number of iterations that are chosen based on visual inspection and experience.

The system matrix $\mathbf{S}$ encodes the applied imaging sequence and can be interpreted in two different ways. It can be interpreted either as a multi-patch dataset, where the entire 3D sequence is split into the individual 2D subsequences, or as a single-patch dataset. A patch refers to a subvolume that is moved by the dynamic selection field and consists of a single full DF cycle. The multi-patch approach is common when the applied selection field changes only in a step-wise fashion, but it can also be applied when the

selection field changes only slowly compared to the rapid drive field-induced movement. This multi-patch reconstruction approach was used in the first generation of our perfusion imager[8] and studied in more detail by ref. 86. Alternatively, one can also interpret the entire imaging sequence as a single-patch dataset, which was considered by ref. 87 for Cartesian 2D trajectories. The multi-patch approach has the potential advantage that it may allow to exploit shift-invariant sub-blocks within the system matrix, which can accelerate operations involving the system matrix $S$ within image reconstruction[45]. On the other hand, the single-patch approach can take field imperfections better into account and can also better prevent spectral leakage, which can be induced by non-periodic external signal contributions. Effectively, the key difference between both approaches is that the Fourier transform is applied to smaller signal snippets in the multi-patch case, while the single-patch case applies the Fourier transform to the entire time signal. Since the single-patch reconstruction is fast enough for our purposes, we use this approach in all experiments shown in this paper. For the system matrix analysis performed in the subsubsection "System matrix analysis" we, however, consider both the multi-patch and the single-patch data interpretation since this gives much deeper insight into the system matrix structure.

**Perfusion image calculation**. To evaluate perfusion experiments in the subsubsection "Dynamic perfusion experiments", the reconstructed data is processed to obtain the time-to-peak (TTP), mean-transit-time (MTT), relative cerebral-blood-flow (rCBF), and relative cerebral-blood-volume (rCBV). Post-processing includes four consecutive steps: (i) time framing, (ii) filtering and offset correction, (iii) threshold masking, and (iv) parameter map calculation. After explaining these post-processing steps, we give the implemented definitions of the mentioned perfusion parameters. The definitions and the post-processing script that handles the data are based on the work by ref. 56, which provides more details.

Step (i) selects the relevant reconstructed data by taking the time frames that include the entire first passage of the administered bolus. It starts with the injection (5 s before bolus appearance), includes the administered bolus and stops after the passing levels to zero again. This step ensures that only relevant data is processed later, as shown in the leftmost column of Fig. 6. In step (ii), an appropriate filter type smooths the data for a more accurate peak detection and noise suppression. We avoided rectangular windows to reject ringing artifacts and selected a low-pass Hann-filter with a window size of ten samples. The Hann-filter is applied voxel-wise on the Fourier-transformed temporal data, which also shifts the concentration offset to zero by excluding the DC component. Afterwards, in step (iii), a threshold mask reduces image noise by excluding any voxels with an intensity lower than 10% of the maximum value. This increases the readability of perfusion maps, by excluding irrelevant regions, e.g., outside (phantom) vessels. A last post-processing step (iv), calculates the mentioned perfusion parameters based on the following definitions, which are also sketched in the top row of Fig. 6. The reconstructed time-dependent solution of the entire volume in the FOV is $\tilde{c} : [0, T] \to \mathbb{R}_+^N$ with $N = n_x \times n_y \times n_z$ voxels. $\tilde{c}(t)$ describes the reconstructed volume of all measured time frames, that record the entire bolus administration.

TTP. We define the time-to-peak (TTP) as the time elapsed between a chosen reference point (the bolus injection $t_0$) and the measured signal maximum of the bolus passing. The $\mathbf{TTP} \in \mathbb{R}^N$ is calculated element-wise for all voxel $n \in \{1, \dots, N\}$ via $\mathrm{TTP}_n = \arg\max_t(\tilde{c}_n(t))$, where $\tilde{c}_n(t)$ is the concentration over time of the $n$th voxel[88].

MTT. With mean-transit-time (MTT), we refer to the measure of the average time interval that a particle or blood cell spends inside an organ or vessel and it strongly correlates with the full width at half maximum (FWHM) of the bolus concentration on passing (for low tissue perfusion)[89,90]. The time interval of the FWHM was therefore selected as the MTT $\in \mathbb{R}^N$ in this work.

rCBF. The relative cerebral-blood-flow (rCBF) equals the highest positive gradient of the concentration over time $\tilde{c}_n(t)$, as in $\mathrm{rCBF}_n = \arg\max_t(\frac{\mathrm{d}}{\mathrm{d}t}\tilde{c}_n(t))$, which is evaluated element-wise for all voxels $n$ to obtain $\mathbf{rCBF} \in \mathbb{R}^N$.

rCBV. We derive the relative cerebral-blood-volume (rCBV) data $\mathbf{rCBV} \in \mathbb{R}^N$ from an element-wise evaluation of the integral (area under the curve (AUC)) of the concentration $\tilde{c}_n(t)$ in the $n$th voxel, over the time interval of the passing bolus ($\tilde{c}_n(t) > 0$) as in $\mathrm{rCBV}_n = \int \tilde{c}_n(t)\,\mathrm{d}t$.

Blood flow and volume are both calculated in a relative manner, due to a missing correct arterial input function (e.g., feeding phantom hose), which poses as a reference by providing the undistorted flow (without perfusion).

## Data availability
Data sets generated during the current study are available from the corresponding author on reasonable request.

## Code availability
The code supporting the findings of this study is publicly available on GitHub. You can access the code repositories through the following links: GUI: https://github.com/MagneticParticleImaging/MPIUI.jl Signal generation and reception: https://github.com/tknopp/RedPitayaDAQServer Hardware and measurement control: https://github.com/MagneticParticleImaging/MPIMeasurements.jl Image reconstruction: https://github.com/MagneticParticleImaging/MPIReco.jl

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

## Acknowledgements

We thank Florian Sevecke for technical support during the realization of numerous scanner components. We further thankfully acknowledge the financial support by the German Research Foundation (DFG, grant numbers KN 1108/7-1 and GR 5287/2-1). We also thank the developers of Makie.jl supporting us by answering technical questions when creating Figs. 3 and 4. Finally, we are grateful to Christian Findeklee for their discussions and insights into resonant circuit decoupling. We acknowledge financial support from the Open Access Publication Fund of UKE - Universitätsklinikum Hamburg-Eppendorf.

## Author contributions

F.T., F.F., F.M., T.K. and M.G. contributed to the system's conceptualization. F.T., F.F. and F.M. constructed the MPI components. N.H. and T.K. developed the software. F.T., F.F., F.M., N.H., M.B., M.M. and T.K. contributed to experiment planning and execution. J.S. provided assistance with the reception system. T.K. and M.G. supervised the project. F.T., F.F., F.M. and T.K. contributed to writing the paper with the support of N.H., M.B. and M.M. All authors reviewed the manuscript.

## Funding

## Competing interests

The authors declare no competing interests.
