## [Peer Review File · Communications Engineering]

Reviewers' comments:

Reviewer #2 (Remarks to the Author):

This manuscript demonstrates the development and use of a 3D human-sized brain MPI scanner. The authors have validated this system using static phantoms as well as phantoms of perfusion. This is the second generation of the human head scanner, which has shown significant improvements including 3D abilities and a frame rate of 4Hz.

This work is important in the fields of both imaging and medicine, where researchers in a variety of fields will be interested in this work. The authors have provided convincing findings, thorough methods and appropriate conclusions.

Overall I think this is a well written, comprehensive and enjoyable manuscript. I have little to comment on regarding the system overview and data processing as this is outside my field of expertise. However, I could follow these sections and, as someone who does not know much about this, could learn some things.

I only have a few minor comments which I think would strengthen the manuscript and help other readers better understand the work.

1. Figure 7a – I believe each reconstruction is adjusted so there is the max/min signal intensities displayed. Would be nice to have one example where the same max/min is applied to all of the different iron amounts to visually see the differences in the signal output for each (ex. 512 ug = bright, with the subsequent amounts dimmer)
2. The spatial resolution was performed using Resotran- this is known to have a poor resolution vs some of the newer MPI-tailored NPs. What about synomag? I don't think this is a required experiment but would be worth discussing more since this could be a potential use if there is clinical approval. How much better resolution would be expected, etc?
3. The total imaging time, and total time including reconstruction would be important to include in these experiments. If possible, I would be interested in seeing time from beginning of scan to image output, for example, in a clinical setting.

I look forward to seeing the revised manuscript

Reviewer #3 (Remarks to the Author):

This is an innovative scientific contribution to the MPI community that develops a human-sized 3D real time MPI scanner for cerebral applications. The researches performed by the authors are important and fundamental. This reviewer just has several minor comments for the authors.

Introduction

1. Portable low-field MRI has been used in clinic, please cite these reference papers.
DOI: 10.1038/s44222-023-00086-w
DOI: 10.1161/STROKEAHA.123.043146

Experiments and Results

2. For the sensitivity study, in Figure 7a, at 4ug iron, why were the samples positioned on the left and center both seen on the right side?
3. The information of the scanning time for each experiment should be provided.
4. For the multi-contrast experiments, please elaborate more on the differences of the two tracers tested in the experiments. Why the channel leakage was only detected in the Synomag channel?
5. Did the author performed any statistical analysis for their data? If so, please include a separate paragraph to summarize the statistical methods and provide the information of the statistical software they used.

Device Safety

6. One limitation of this study is that only phantoms were included for the experiments, it is this reviewer's suggestion that the author should discuss on the limitations of this study, such as the animal studies were not included for the device safety tests.

SYSTEM CHARACTERIZATION OF A HUMAN-SIZED 3D REAL-TIME MAGNETIC PARTICLE IMAGING SCANNER FOR CEREBRAL APPLICATIONS

FLORIAN THIEBEN, FYNN FOERGER, FABIAN MOHN, NIKLAS HACKELBERG, MARIJA BOBERG, JAN-PHILIPP SCHEEL, MARTIN MÖDDEL, MATTHIAS GRAESER, AND TOBIAS KNOPP

GENERAL COMMENTS

Many thanks to all reviewers for taking the time to assess our manuscript carefully. The comments for improving our manuscript were very helpful. In this reply we outline the changes that we made to the manuscript and answer the issues raised by the reviewers point by point. We shortened the abstract down to meet the journal guidelines. Further, we changed one nomenclature. The term "High Current Circuit (HCC)" has been replaced with "High Current Resonator (HCR)" to better reflect the purpose of this circuit.

REVIEWER #2

This manuscript demonstrates the development and use of a 3D human-sized brain MPI scanner. The authors have validated this system using static phantoms as well as phantoms of perfusion. This is the second generation of the human head scanner, which has shown significant improvements including 3D abilities and a frame rate of 4Hz.

This work is important in the fields of both imaging and medicine, where researchers in a variety of fields will be interested in this work. The authors have provided convincing findings, thorough methods and appropriate conclusions.

Overall I think this is a well written, comprehensive and enjoyable manuscript. I have little to comment on regarding the system overview and data processing as this is outside my field of expertise. However, I could follow these sections and, as someone who does not know much about this, could learn some things.

I only have a few minor comments which I think would strengthen the manuscript and help other readers better understand the work.

Answer: We thank you very much for this assessment of our work.

FIGURE 1. Concentration visualization. The first row contains reconstructed images with individual maximum and minimum values per concentration. In the second row a global maximum/minimum per iron amount is set.

Specific aspects.

1. Figure 7a – I believe each reconstruction is adjusted so there is the max/min signal intensities displayed. Would be nice to have one example where the same max/min is applied to all of the different iron amounts to visually see the differences in the signal output for each (ex. 512 ug = bright, with the subsequent amounts dimmer)

Answer: In Figure 1 we prepared such a plot with local and global signal maximum and minimum. However, due to the logarithmic concentration series, all images with a global maximum/minimum and with iron amounts below 128 μg would not provide any visual information. From our point of view, this type of visualization does not add enough useful information to justify an additional row in Figure 7a.

2. The spatial resolution was performed using Resotran- this is known to have a poor resolution vs some of the newer MPI-tailored NPs. What about synomag? I don't think this is a required experiment but would be worth discussing more since this could be a potential use if there is clinical approval. How much better resolution would be expected, etc?

Answer: We agree that synomag most likely would provide better sensitivity and spatial resolution. Resotran and Resovist are basically ferucarbutran. In [5], a 2.2 fold better signal response was measured for synomag compared to Resovist. The improvement in spatial resolution has so far only been hinted at in the literature but not yet quantified [8]. The achievable resolution depends not only on the particles themselves, but also on reconstruction parameters and algorithms. It is therefore difficult to estimate the achievable resolution of synomag without carrying out an experiment.

In this work, we focused on a clinically approved tracer, since this will realistically be chosen in first human trials in the upcoming years. We think it makes most sense to not generate better pictures/sensitivities with an advanced tracer at this point when the next step will be the switch to human trials using Resotran. In the outlook, we indicated the superiority of MPI tailored MNPs like synomag but did not comment

on specific advantages like better sensitivity and spatial resolution. We changed the manuscript to make this more clear:

Revised: Tailored MPI tracers, such as *perimag* and *synomag*, with future clinical approvals, promise to increase system performance in sensitivity and spatial resolution and further expand the range of applications.

3. The total imaging time, and total time including reconstruction would be important to include in these experiments. If possible, I would be interested in seeing time from beginning of scan to image output, for example, in a clinical setting.

Answer: This is a very good point, all measurements were performed without frame averaging, so with a 4 Hz frame rate. For the results shown, a detailed evaluation of the measurements was carried out, thus reconstructions were performed retrospectively.

Imaging latency of the MPI scanner can be divided into two aspects. First signal acquisition and second image reconstruction and visualization. On the RedPitaya, signal acquisition takes place simultaneously with signal generation. Acquired data is processed and sent to the workstation. On the workstation, the acquired signal is collected and the raw-data is stored for further processing. Reconstructions can be performed retroactively on the stored data or based on the received signal stream. The reconstruction runtime depends on reconstruction parameters like the number of iterations and the utilized optimization algorithm. As a final step, the reconstructed image is visualized. Measuring the end-to-end signal processing latency is challenging and the reconstruction runtime depends on the application. However, using our GUI *MPIUI.jl*, we are able to perform online reconstructions with a latency of less than one second. The GUI features room for more latency reduction, such as accessing a frame earlier in the processing pipeline, which have not yet been exploited. We now added the measurement time to all experiments and revised the discussion to include the imaging latency.

Revised: Reconstructions of the presented results were performed retrospectively on the stored measurements. However, the operational control and data processing software makes it possible to reconstruct online. Although the software is not yet optimized for this use case, the GUI is able to perform online reconstruction with a latency of less than one second between signal acquisition and image visualization. Further latency reduction could be achieved by accessing the measurement data earlier in the processing pipeline and by adapting additional methods from Knopp et al [4].

REVIEWER #3

This is an innovative scientific contribution to the MPI community that develops a human-sized 3D real time MPI scanner for cerebral applications. The researches performed by the authors are important and fundamental. This reviewer just has several minor comments for the authors.

Answer: Thank you very much for this assessment of our work.

Introduction.

1. Portable low-field MRI has been used in clinic, please cite these reference papers. DOI: 10.1038/s44222-023-00086-w DOI: 10.1161/STROKEAHA.123.043146

Answer: This is a very good point, portable devices using non-ionizing radiation, within the clinics is a good aspect to include in the introduction.

Revised: Recently, portable Magnetic Resonance Imaging (MRI) systems with low B_0 fields proved the potential of bedside brain imaging devices in the intensive care unit (ICU) [3, 6].

Experiments and Results.

1. For the sensitivity study, in Figure 7a, at 4 μ g iron, why were the samples positioned on the left and center both seen on the right side?

Answer: Thank you for pointing this out. In contrast to other imaging modalities, noise in MPI can lead to structured image artifacts. For low iron amounts superimposed noise can exceed the MPNs signal response. In combination with strong regularization parameters for the reconstruction algorithm, structured image artifacts may occur. These artifacts can be misinterpreted to be tracer signal. Therefore, Greaser et al. suggested to perform sensitivity studies at multiple positions [2]. The positions were chosen to ensure that the spatial position of the reconstruction is caused by a sample at the same position and not by a structured artifact. We clarify this as follows:

Revised: In terms of sensitivity, a dilution series measured at different positions was required. In contrast to other imaging modalities, noise in MPI can lead to structured image artifacts. For low iron amounts, superimposed noise in combination with strong regularization parameters for the reconstruction algorithm, structured image artifacts may occur [2].

Answer: In the original manuscript we did not explicitly state that 4 μ g iron sample did only contain artifacts. To make this more clear, we state this now explicitly:

Revised: At 4 μ g iron, a blurred dot can be seen, but its position does not change for different sample position indicating that the seen dot is caused by system background and not by the sample itself. Thus, the detection limit of the scanner is reached at 8 μ g iron.

2. The information of the scanning time for each experiment should be provided.

Answer: This is a very good point, all shown reconstructions were performed without frame averaging, so at 4 Hz frame rate. In this work the measurements were reconstructed afterwards. We now added this statement to all corresponding experiments.

3. For the multi-contrast experiments, please elaborate more on the differences of the two tracers tested in the experiments. Why the channel leakage was only detected in the Synomag channel?

Answer: To our knowledge, neither the multi-contrast performance of specific tracer pairs, nor the underlying phenomena for the observed channel leakage yet has been

investigated. However, asymmetric occurrence of channel leakage has been observed in the first work on this topic [7]. Regarding multi-contrast, our focus was on demonstrating the capability of the MPI scanner. To make clear that the investigation of channel leakage is not in the focus of our work, we revised the discussion as follows.

Revised: We observe minor channel leakage in the multi-contrast tomogram, which is typical for multi-contrast imaging [7]. In general, the underlying phenomena for the observed channel leakage in multi-contrast imaging needs to be further investigated.

4. Did the author performed any statistical analysis for their data? If so, please include a separate paragraph to summarize the statistical methods and provide the information of the statistical software they used.

Answer: All experiments in this work have been performed at least twice to ensure reproducibility. Qualitatively, no significant difference could be found. A statistical evaluation of the scanner performance is still pending and planned in future work. The system-characterizing values from the experiments are specified without accuracy estimation, but provide a good estimate of the corresponding order of magnitude. We added the following paragraph to make this clear:

Revised: All experiments in this work have been performed at least twice to ensure reproducibility. Qualitatively, no significant difference could be found. The system-characterizing values obtained from the experiments are given without an accuracy estimate, but provide a good estimate of the corresponding order of magnitude. Future applications can use the high frame rate of the scanner to aggregate and statistically evaluate the acquired data.

Device Safety.

1. One limitation of this study is that only phantoms were included for the experiments, it is this reviewer's suggestion that the author should discuss on the limitations of this study, such as the animal studies were not included for the device safety tests.

Answer: Thank you for pointing that out. We agree that we cannot draw any conclusions about the safety of animals or humans from phantom experiments. Device safety was not our main focus of this work. Currently, we are working on a dedicated analysis. We now addressed this issue in the discussion.

Revised: It should be noted that only phantoms were used in the studies presented. However, the corresponding measurement parameters were carefully chosen to be reasonable for future studies in humans. Ultimately, conclusive assertions concerning safety and tolerability in humans can only be made after animal or human studies [1].

REFERENCES

- [1] Caroline Billings, Mitchell Langley, Gavin Warrington, Farzin Mashali, and Jacqueline Anne Johnson. Magnetic Particle Imaging: Current and Future Applications, Magnetic Nanoparticle Synthesis Methods and Safety Measures. *International Journal of Molecular Sciences*, 22(14):7651, July 2021.

- [2] Matthias Graeser, Tobias Knopp, Patryk Szwargulski, Thomas Friedrich, Anselm Von Gladiss, Michael Kaul, Kannan M. Krishnan, Harald Ittrich, Gerhard Adam, and Thorsten M. Buzug. Towards Picogram Detection of Superparamagnetic Iron-Oxide Particles Using a Gradiometric Receive Coil. *Scientific Reports*, 7(1):6872, 2017.
- [3] W. Taylor Kimberly, Annabel J. Sorby-Adams, Andrew G. Webb, Ed X. Wu, Rachel Beekman, Ritvij Bowry, Steven J. Schiff, Adam De Havenon, Francis X. Shen, Gordon Sze, Pamela Schaefer, Juan Eugenio Iglesias, Matthew S. Rosen, and Kevin N. Sheth. Brain imaging with portable low-field MRI. *Nature Reviews Bioengineering*, 1(9):617–630, July 2023.
- [4] T. Knopp and M. Hofmann. Online reconstruction of 3D magnetic particle imaging data. *Physics in Medicine & Biology*, 61(11):N257–N267, 2016.
- [5] Peter Ludewig, Matthias Graeser, Nils D. Forkert, Florian Thieben, Javier Rández-Garbayo, Johanna Rieckhoff, Katrin Lessmann, Fynn Förger, Patryk Szwargulski, Tim Magnus, and Tobias Knopp. Magnetic particle imaging for assessment of cerebral perfusion and ischemia. *WIREs Nanomedicine and Nanobiotechnology*, 14(1), January 2022.
- [6] Mercy H. Mazurek, Nethra R. Parasuram, Teng J. Peng, Rachel Beekman, Vineetha Yadlapalli, Annabel J. Sorby-Adams, Dheeraj Lalwani, Julia Zabinska, Emily J. Gilmore, Nils H. Petersen, Guido J. Falcone, Nanthiya Sujjantararat, Charles Matouk, Sam Payabvash, Gordon Sze, Steven J. Schiff, Juan Eugenio Iglesias, Matthew S. Rosen, Adam De Havenon, W. Taylor Kimberly, and Kevin N. Sheth. Detection of Intracerebral Hemorrhage Using Low-Field, Portable Magnetic Resonance Imaging in Patients With Stroke. *Stroke*, 54(11):2832–2841, November 2023.
- [7] J Rahmer, A Halkola, B Gleich, I Schmale, and J Borgert. First experimental evidence of the feasibility of multi-color magnetic particle imaging. *Physics in Medicine & Biology*, 60(5):1775, 2015.
- [8] Patrick Vogel, Thomas Kampf, Martin Rückert, Cordula Grüttner, Anja Kowalski, Henrik Teller, and Volker Behr. Synomag®: The new high-performance tracer for magnetic particle imaging. *International Journal on Magnetic Particle Imaging IJMPI*, 7(1), 2021.

REVIEWERS' COMMENTS:

Reviewer #2 (Remarks to the Author):

Recommend manuscript is accepted

Reviewer #3 (Remarks to the Author):

The Authors have addressed all of my concerns with the original manuscript. The revised manuscript is acceptable for publication from my point of view.